# Single cell analyses reveal distinct adaptation of typhoidal and non-typhoidal *Salmonella enterica* serovars to intracellular lifestyle

Tatjana Reuter[1], Felix Scharte[1], Rico Franzkoch[1,2], Viktoria Liss[1,2], Michael Hensel[1,3]*

1 Abt. Mikrobiologie, Universität Osnabrück, Osnabrück, Germany, 2 iBiOs–integrated Bioimaging Facility Osnabrück, Universität Osnabrück, Osnabrück, Germany, 3 CellNanOs–Center of Cellular Nanoanalytics Osnabrück, Universität Osnabrück, Osnabrück, Germany

☯ These authors contributed equally to this work.
* Michael.Hensel@uni-osnabrueck.de

**Data Availability Statement:** All relevant data are within the manuscript and its Supporting Information files. Original data files are available at

## Abstract

*Salmonella enterica* is a common foodborne, facultative intracellular enteropathogen. Human-restricted typhoidal *S. enterica* serovars Typhi (STY) or Paratyphi A (SPA) cause severe typhoid or paratyphoid fever, while many *S. enterica* serovar Typhimurium (STM) strains have a broad host range and in human hosts usually lead to a self-limiting gastroenteritis. Due to restriction of STY and SPA to primate hosts, experimental systems for studying the pathogenesis of typhoid and paratyphoid fever are limited. Therefore, STM infection of susceptible mice is commonly considered as model system for studying these diseases. The type III secretion system encoded by *Salmonella* pathogenicity island 2 (SPI2-T3SS) is a key factor for intracellular survival of *Salmonella*. Inside host cells, the pathogen resides within the *Salmonella*-containing vacuole (SCV) and induces tubular structures extending from the SCV, termed *Salmonella*-induced filaments (SIF). This study applies single cell analyses approaches, which are flow cytometry of *Salmonella* harboring dual fluorescent protein reporters, effector translocation, and correlative light and electron microscopy to investigate the fate and activities of intracellular STY and SPA. The SPI2-T3SS of STY and SPA is functional in translocation of effector proteins, SCV and SIF formation. However, only a low proportion of intracellular STY and SPA are actively deploying SPI2-T3SS and STY and SPA exhibited a rapid decline of protein biosynthesis upon experimental induction. A role of SPI2-T3SS for proliferation of STY and SPA in epithelial cells was observed, but not for survival or proliferation in phagocytic host cells. Our results indicate that reduced intracellular activities are factors of the stealth strategy of STY and SPA and facilitate systemic spread and persistence of the typhoidal *Salmonella*.

## Author summary

Typhoidal *Salmonella enterica* serovars Typhi (STY) and Paratyphi A (SPA) cause a major disease burden to the human population. The restriction of these pathogens to human

the Zenodo repository under DOI 10.5281/zenodo.
4898834.

**Funding:** This work was supported by BMBF grant
031L0093A to MH within the Infect-ERA
consortium SalHostTrop. Further funding was
granted by the DFG through SFB944 to MH. The
funders had no role in study design, data collection
and analysis, decision to publish, or preparation of
the manuscript.

**Competing interests:** The authors have declared
that no competing interests exist.

hosts limits experimental analyses of molecular mechanisms of diseases. *S. enterica* sero-
var Typhimurium is commonly used as surrogate model for typhoidal *Salmonella* (TS),
and allowed the identification of virulence factors for intracellular lifestyle of *S. enterica* in
mammalian host cells. If virulence factors, such as the *Salmonella* Pathogenicity Island 2-
encoded type III secretion system (SPI2-T3SS) have similar roles for intracellular lifestyle
of TS is largely unknown. We analyzed, on single cell level, the intracellular activities of
STY and SPA in comparison to STM. STY and SPA deploy SPI2-T3SS to actively manipu-
late their host cells, but with far lower frequency than STM. Our work supports a model of
TS as stealth pathogens that persist in host cells.

## Introduction

*Salmonella enterica* is a versatile gastrointestinal pathogen with the ability to cause diseases
ranging from acute, usually self-limiting gastroenteritis due to infections by non-typhoidal *Sal-
monella* (NTS) to severe systemic infections caused by typhoidal *Salmonella* (TS) serovars.
Infections by *S. enterica* serovars such as Typhi (STY) and Paratyphi A (SPA) represent a con-
tinuing threat to human health. Particular in countries with low standards of sanitation, TS
infections are endemic. The global burden of disease by TS infections is continuously high
with about 27,000,000 infected people and 200,000 deaths annually worldwide, and the
increased frequency of multidrug-resistant strains of TS, as well as coinfections cause problems
for treatment of typhoid fever [1,2].

While NTS infections are commonly associated with a strong inflammatory response lead-
ing to effective immune defense at the intestinal epithelium, TS infections lack this response
and allow the pathogen to enter circulation and lymphatic system, and ultimately infect solid
organs (reviewed in [3]). The ability to survive phagocytosis and to persist and proliferate in
infected host cells is considered a key virulence trait of *S. enterica* [4].

The restriction of STY and SPA to primate host organisms limits experimental approaches
to study pathogenesis of typhoid and paratyphoid fever. Infection of susceptible mice by *S.
enterica* sv. Typhimurium (STM) is commonly considered as model system for studying the
diseases caused by highly human-adapted STY and SPA. Important virulence traits of *S. enter-
ica* are also investigated on the cellular level using various cell culture systems for infection.

The type III secretion system encoded by *Salmonella* Pathogenicity Island 2 (SPI2-T3SS) is
of central importance for systemic virulence of STM in a murine infection model, as well as for
intracellular proliferation in cellular infection models with human or murine cells. Intracellu-
lar *Salmonella* within the SCV deploy the SPI2-T3SS to translocate into host cells a cocktail of
more than 30 effector proteins. These effector proteins manipulate various host cell functions
such as vesicular transport, the actin and microtubule cytoskeleton, ubiquitination, apoptosis,
releases of cytokines, antigen presentation by MHCII, and many more. Collectively, SPI2-T3SS
effector proteins are required for intracellular survival and replication and the proliferation
within host tissues. A remarkable difference between STM, STY and SPA is lack of a large
number of SPI2-T3SS effector proteins in TS (reviewed in [5]). This is due to absence of the
virulence plasmid and bacteriophage genomes harboring effector genes, and the pseudogen-
ization of effector genes. Thus, function of the SPI2-T3SS and related phenotypes may be
absent or altered in cells infected by STY or SPA. If findings on the role of SPI2-T3SS made in
STM are also applicable to virulence of STY and SPA has not been investigated. A prior study
reported that the SPI2-T3SS is not required for the intracellular survival and replication in the
human macrophage cell line THP-1 which indicates that the SPI2 is not as important for STY

as for STM [6]. A recent genome-wide Tn-Seq screen for STY in a humanized mouse model did not indicate a role of SPI2-T3SS within infection for 24 h [7]. Other studies reported requirement for SPI2-T3SS in release of typhoid toxin by cells infected by STY [8].

We set out to investigate the cellular microbiology of STY and SPA with focus on function of SPI2-T3SS and related phenotypes. To analyze the response of intracellular STM, STY or SPA to the host cell environment, we applied single cell-based approaches, such as fluorescence microscopy to follow translocation of effector protein, and correlative light and electron microscopy (CLEM) for ultrastructural analyses of the host cell endosomal system [9], and potential manipulation by intracellular *S. enterica*. The fate of individual intracellular *S. enterica* is highly divergent [10–12], thus we utilized various dual fluorescence reporters to determine the dynamic formation of distinct populations on single cell level.

Our analyses demonstrate that the SPI2-T3SS is functional in STY and SPA and mediates endosomal remodeling similar to STM. However, this manipulation of the host cell is only observed in a small number of host cells, and the frequency of STY and SPA showing intracellular activities is much lower compared to STM. The parsimonious deployment of virulence factors such as SPI2-T3SS can be considered as part of the stealth strategy of TS and may contribute to evade host immune responses.

## Results

### Genes encoding the SPI2-T3SS are induced in intracellular SPA and STY

To investigate the response of STY and SPA to the intracellular environment in various host cells, we analyzed if genes in SPI2 encoding the T3SS are expressed. For this, we used flow cytometry (FC) of strains harboring a dual fluorescent protein (FP) reporter plasmid with constitutive expression of DsRed for gating of intracellular *Salmonella*, and a fusion of the promoter of *ssaG* sfGFP to determine induction of a representative operon encoding SPI2-T3SS subunits. Analyses were performed in WT and Δ*ssrB* strains of STM, SPA or STY. Because *ssrB* is a local transcriptional regulator of genes encoding the SPI2-T3SS, we anticipated highly reduced activation of P$_{ssaG}$::sfGFP in Δ*ssrB* background [13,14]. The functionality of the reporter was tested *in vitro* (**Fig 1A**). In LB broth and minimal media with high phosphate concentration and neutral pH, referred to as PCN (25), pH 7.4, there was no induction of P$_{ssaG}$::sfGFP in STY and SPA, but slight induction in STM. In SPI2-inducing minimal media with a low phosphate concentration referred to as PCN (0.4), pH 5.8, P$_{ssaG}$ induction was detected in WT strains of the three serovars, while no induction was detected in strains with deletion of *ssrB*. While STM, STY and SPA all activated SPI2 gene expression in a SsrB-dependent manner upon growth under SPI2-inducing conditions, the levels of expression were different. As indicated by X-median values for sfGFP intensities, much higher sfGFP levels were detected for STM than for STY and SPA after growth in PCN-P. Furthermore, STY and SPA exhibited higher heterogeneity with subpopulations of non-induced and weakly induced cells. In contrast, P$_{ssaG}$::sfGFP was induced more homogeneously in almost all STM in PCN-P manner (**Fig 1A**). The function of reporter P$_{ssaG}$::sfGFP and the role of SsrB in controlling SPI2 promoters in intracellular *Salmonella* was further corroborated by microscopy of STM-, STY- or SPA-infected host cells. Representative micrographs for HeLa infection show the expression and absence of expression in WT or Δ*ssrB* background, respectively (**S1 Fig**).

We next determined P$_{ssaG}$ induction by *Salmonella* in various host cell lines that are frequently used to study cellular microbiology of *Salmonella* infections. As host cells, we deployed non-polarized human epithelial cell line HeLa (**Fig 1B**), human macrophage cell line U937 (**Fig 1C**), and murine macrophage cell line RAW264.7 (**Fig 1D**). In addition, human primary macrophages (**Fig 1E**) were infected. In none of these host cells, *ssrB* mutant strains showed

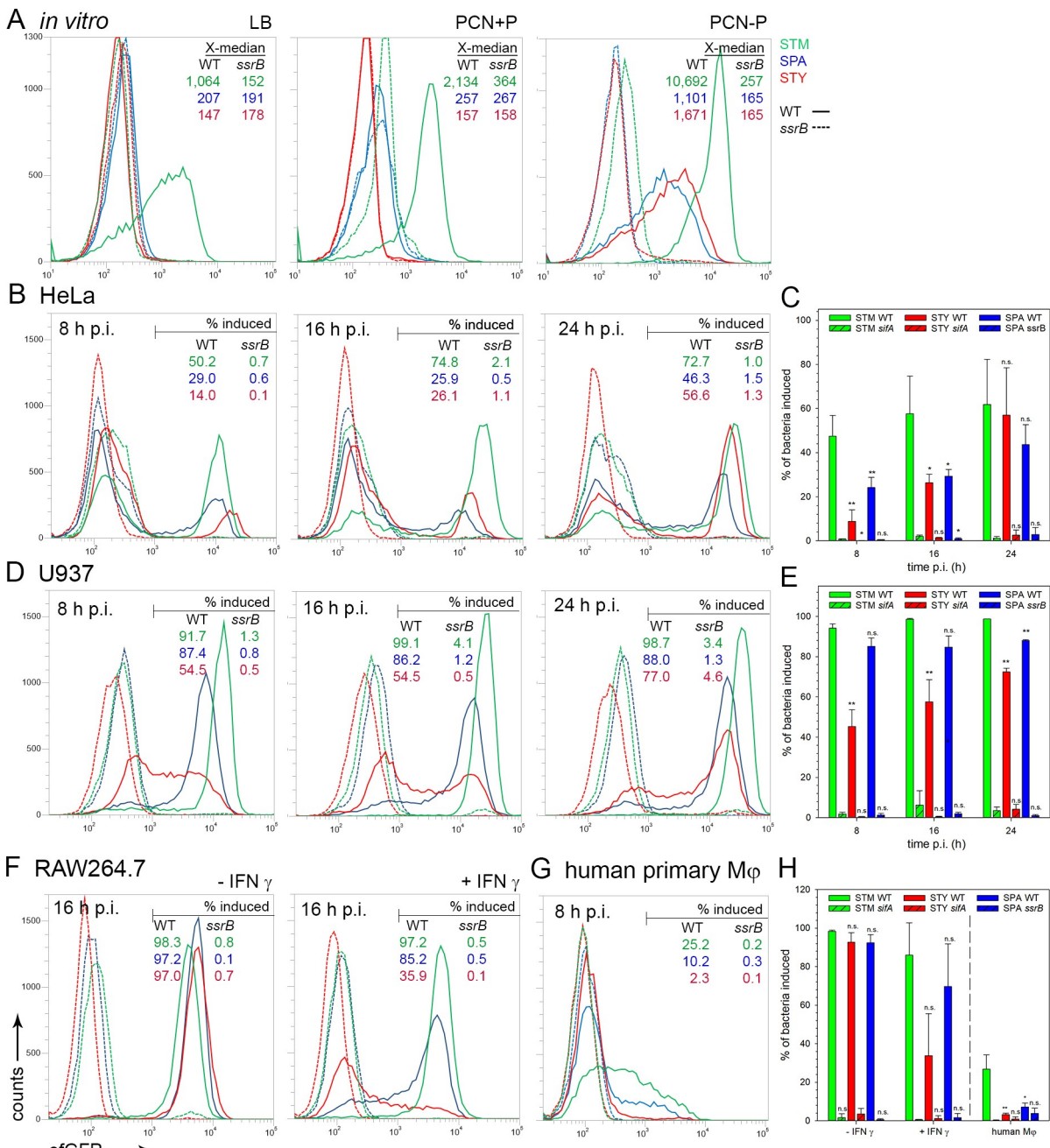

**Fig 1. Expression of genes encoding the SPI2-T3SS by intracellular STM, SPA and STY.** STM (green), SPA (blue), or STY (red) WT (solid lines) or *ssrB* mutant (dashed lines) strains were used, all harboring plasmid p3776 for constitutive expression of DsRed, and sfGFP under control of the *ssaG* promoter. **A)** Various strains were grown in non-inducing PCN medium (PCN pH 7.4, 25 mM $P_i$), or inducing PCN medium (PCN pH 5.8, 0.4 mM $P_i$). After culture o/n with aeration by rotation at 60 rpm in a roller drum, aliquots of cultures were fixed, and bacteria were analyzed by flow cytometry (FC). The X-median values for the sfGFP-positive bacteria are indicated. Various strains were used to infect HeLa cells (**B, C**), U937 macrophages (**D, E**), RAW264.7 macrophages (**F, H**) without or with activation by γ-interferon (IFN-γ), or human primary macrophages (**G**). At 8 h, 16 h, or 24 h p.i. as indicated, host cells were lysed in order to release bacteria. For FC, at least 10,000 bacteria-sized particles with DsRed fluorescence were gated and GFP intensities were quantified. The percentage of intracellular bacteria with induction of P$_{ssaG}$::sfGFP is indicated for WT and Δ*ssrB* strains. The data sets in **A)**, **B)**, **D)**, **F)**, and **G)** show data representative for three independent experiments with similar outcome. Means and standard deviations of three biological replicates of infection of HeLa (**C**), U937 (**E**), or RAW264.7 cells (**H**) are shown. Statistical analysis was performed by One-way ANOVA and are indicated as follows: n.s., not significant; *, P< 0.05; **, P< 0.01; ***, P< 0.001.

P$_{ssaG}$ induction, while induction was observed in STM, STY and SPA WT strains. In HeLa cells, the three serovars showed highest sfGFP intensities at 24 h p.i., with STM always showing higher sfGFP intensities than STY or SPA. In U937 cells, STM again showed the highest sfGFP levels at 8 h, 16 h and 24 h p.i. While the intracellular population of STM and SPA showed rather uniform induction, a heterogeneous response was observed for intracellular STY, with single cells showing various levels ranging from very low to high sfGFP intensities. At 24 h p.i., the distribution of the STY population changed, and sfGFP intensity increased. In the murine macrophage-like cell line RAW264.7 at 16 h p.i., the sfGFP intensities were similar for STM, STY and SPA. Activation by interferon γ (IFN-γ) increases antimicrobial activities of macrophages, and we investigated the effect of activation on reporter expression using RAW264.7 macrophages as infection model. After activation of RAW264.7 by IFN-γ STY only produced very low sfGFP fluorescence, SPA showed higher intensities, and highest signal intensities were observed for STM (**Fig 1F**). In human primary macrophages, only a part of the population of SPA and STM WT showed higher sfGFP levels than the ΔssrB strains (**Fig 1G**). Human primary macrophages used in this study were predominantly M1-polarized, and the high antimicrobial activity of these phagocytes likely limits the intracellular activities of STM, STY and SPA.

As a complementary approach, we analyzed U937 cells infected by STM, STY or SPA for induction of the P$_{ssaG}$::sfGFP reporter (**Fig 2**). While the majority of intracellular STM showed induction of P$_{ssaG}$::sfGFP at 8 h p.i., a lower frequency was observed for STY and SPA under these conditions. Some U937 cells only contained P$_{ssaG}$-negative STY or SPA, while a larger number of host cells harbored mixtures of P$_{ssaG}$-positive and P$_{ssaG}$-negative bacteria.

## The SPI2-encoded type III secretion system is functional in STY and SPA

Mutant strains of STM deficient in SPI2-T3SS function are highly attenuated in murine models of systemic disease, as well as in intracellular survival and proliferation in various types of murine and human host cells [13,15,16]. Genes encoding the SPI2-T3SS are present in STY and SPA and are not affected by pseudogenization. However, the functionality of SPI2-T3SS in STY and SPA has not been demonstrated experimentally.

We investigated the translocation of SPI2-T3SS effector proteins by intracellular STY or SPA (**Fig 3**). For this, mutant strains deficient in the core components of SPI2-T3SS were generated by Red-mediated mutagenesis. Mutations in ssaV, ssaK or ssaR all ablate effector translocation by the SPI2-T3SS. Because antisera against effector proteins of the SPI2-T3SS are limited in availability and quality, we introduced low copy number plasmids for the expression of alleles of representative effector proteins SseF, SseJ, PipB2, and SseL tagged with HA or M45 epitope tags. Prior work reported the translocation of these tagged effector proteins by STM, and their association with late endosomal/lysosomal host cell membranes [17–20].

We observed that effector proteins were translocated into HeLa cells infected by STY or SPA WT (**Fig 3A** and **3B**). No signals for translocated effector proteins were detected after infection with SPI2-T3SS-deficient strains, as shown for representative STY ΔssaK [sseF::M45] (**Fig 3C**). The signal intensities of effectors translocated by STY varied, with stronger accumulation of SseF and SseL, while signals for SseJ were rather weak. After translocation by SPA, signals of similar high intensities were detected. The amounts of effector proteins detected in STY- or SPA-infected host cells were lower than for STM-infected host cells. The translocation of representative SPI2-T3SS effector proteins SseF, SseJ, PipB2, and SseL by intracellular STM, STY, and SPA was quantified by integration of immunofluorescence signal intensities. The average fluorescence levels for effector proteins detected in STY- or SPA-infected were lower than for STM-infected HeLa cells (**Fig 3D**).

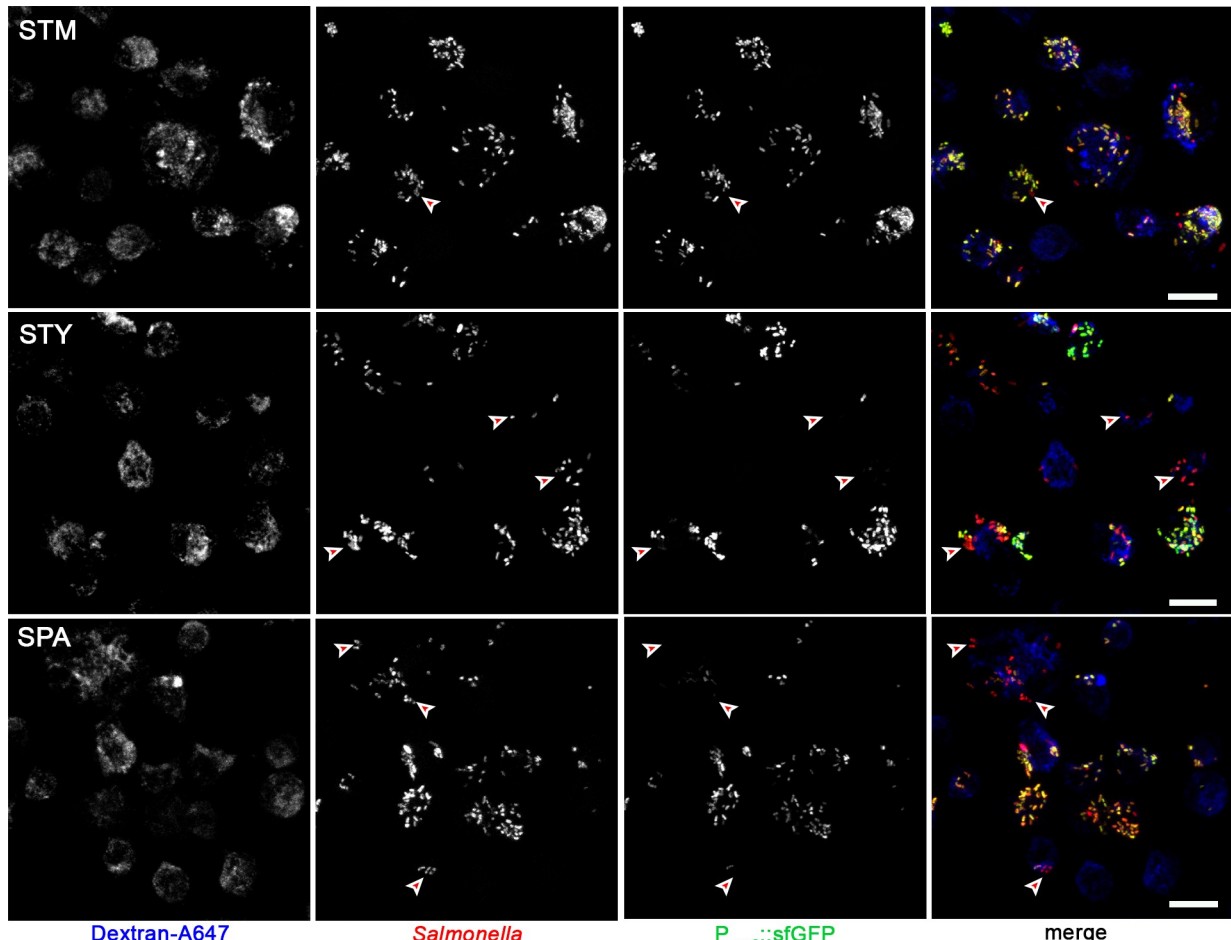

**Fig 2. Heterogeneity of P$_{ssaG}$ induction by intracellular STM, SPA and STY.** U937 cells were infected at MOI 50 with STM, STY, or SPA WT strains all harboring plasmid p3776 for constitutive expression of RFP (red), and sfGFP (green) under control of the *ssaG* promoter. To label the endosomal system, cells were pulsed with dextran-Alexa647 (dextran-A647, blue) from 1 h p.i. until fixation. Cells were fixed with PFA 8 h p.i. and microscopy was performed by CLSM on a Leica SP5 using a 40x objective. Red arrowheads indicate representative, P$_{ssaG}$::sfGFP-negative intracellular *Salmonella*. Scale bars: 10 μm.

We conclude that the SPI2-T3SS is functional in translocation of effector proteins by intracellular STY and SPA. STY and SPA are capable to translocate episomal encoded SseJ of STM, while *sseJ* is a pseudogene in these serovars (Jennings *et al.*, 2017). However, a lower proportion of intracellular STY and SPA compared to STM actively translocate SPI2-T3SS effector proteins.

## SPA and STY inhabit an SCV in infected host cells and remodel the endosomal system

Several cell culture infection models previously demonstrated that STM cells inhabit individual SCV in epithelial or macrophages cell lines, and in primary macrophages [9,21]. A representative marker for this compartment is LAMP1, a member of the family of lysosomal glycoproteins that are membrane integral in late endosomal and lysosomal membranes. Previous work described that each intracellular bacterium is completely enclosed by LAMP1-positive membranes, resulting in individual pathogen-containing compartments [22]. We investigated the formation of SCV in HeLa LAMP1-GFP (**Fig 4**) or RAW264.7 LAMP1-GFP cells (**S2 Fig**) infected with STM, STY, or SPA.

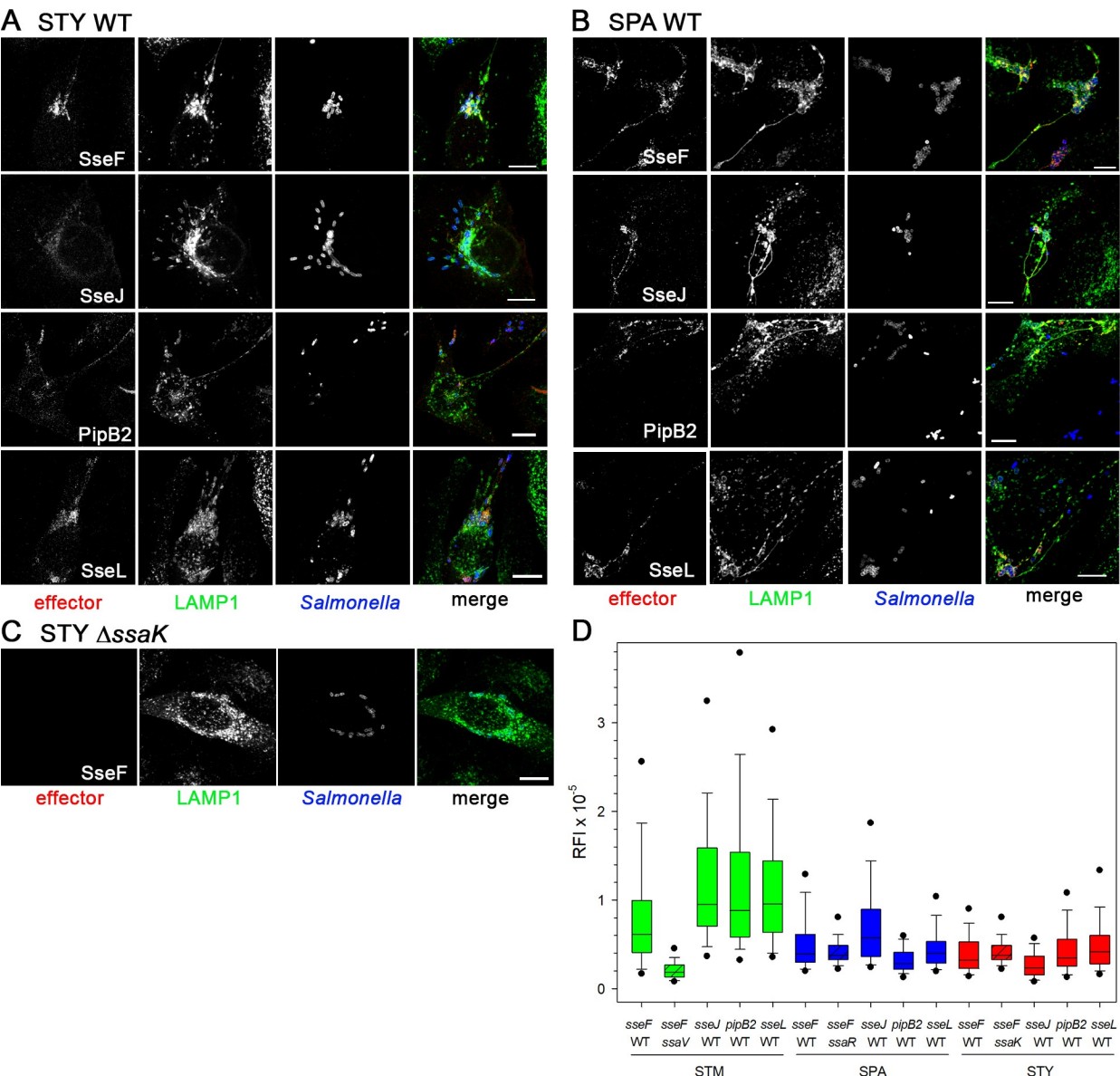

**Fig 3. The SPI2-T3SS of SPA and STY is functional in translocation of effector proteins.** HeLa cells constitutively expressing LAMP1-GFP (green) were infected with STY WT (**A**) or SPA WT (**B**) harboring plasmids for expression of *sseF*::M45, *sseJ*::M45, *pipB2*::M45, or *sseL*::HA as indicated. At 9 h p.i., cells were fixed, permeabilized by Saponin and immunolabeled for M45 or HA epitope tags (red) to localize translocated effector proteins, and with serovar-specific anti O-Ag antibodies to label bacteria (blue). **C**) Micrographs of cells infected with SPI2-T3SS-defcient STY strain Δ*ssaK* harboring a plasmid for expression of *sseF*::M45. Microscopy was performed by CLSM using a Leica SP5 with 100x objective. Scale bars: 10 μm. **D**) Quantification of SPI2-T3SS effector protein translocation by intracellular STM, STY or SPA. HeLa cells were infected as for **A**)-**C**) with STM, STY, or SPA WT strains all harboring plasmids encoding epitope-tagged effector proteins *sseF*, *sseJ*, *pipB2*, or *sseL*, or SPI2-T3SS-deficient strains STM Δ*ssaV*, STY Δ*ssaK*, SPA Δ*ssaR* harboring a plasmid encoding *sseF*::M45. Cells were fixed 8 h p.i., permeabilized, and immunostaining of *Salmonella* LPS and epitope tags of translocated effector proteins was performed. Cells harboring *Salmonella* were randomly selected, and for at least 100 infected cells for each condition, fluorescence signals of translocated effector were measured within ROIs defined by the host cell perimeters and expressed as relative fluorescence intensities (RFI). Boxes define 25th and 75th percentiles. Outlyers in the 5[th] and 95[th] percentiles are indicated by dots. Medians and standard deviations are indicated by lines in boxes and whiskers, respectively.

The massive reorganization of the endosomal system was observed in mammalian host cells infected by STM. One consequence of this reorganization is the formation of extensive tubular vesicles. *Salmonella*-induced filaments (SIF) are tubular vesicles characterized by the

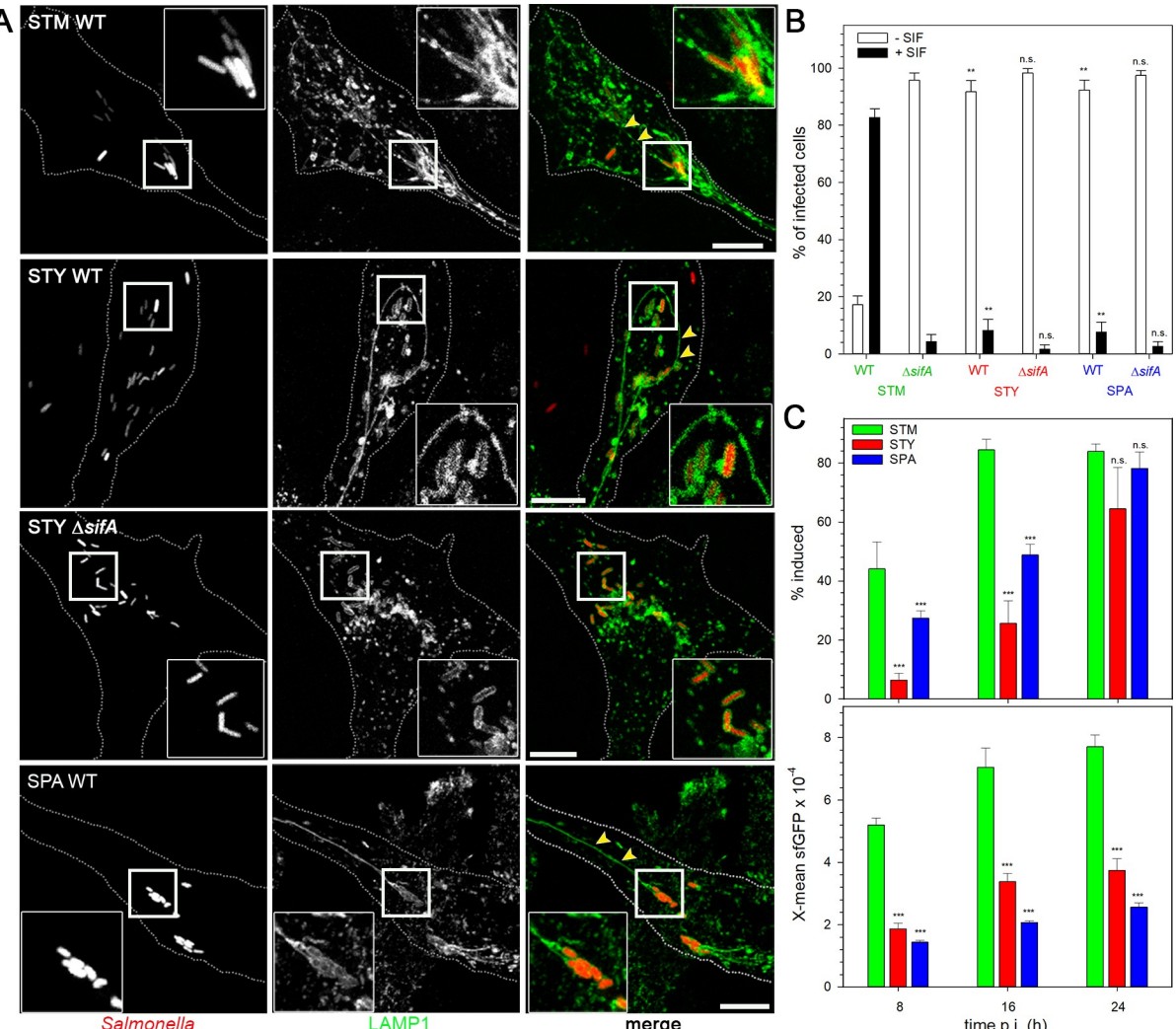

**Fig 4. Endosomal remodeling in host cells infected with STM, SPA, or STY is dependent on function of SifA. A)** HeLa LAMP1-GFP cells were infected with WT or Δ*sifA* strains of STM, STY or SPA as indicated, constitutively expression mCherry (red) at MOI 50. Cells were fixed at 8 h p.i. and microscopy was performed by CLSM on a Leica SP5 using a 100x objective. Representative infected host cells are shown. Boxes indicate areas containing SCV that are magnified in insets. Yellow arrowheads indicate *Salmonella*-induced filaments (SIF). Scale bar: 10 μm. **B)** Quantification of SIF formation in HeLa cell infected by STM, STY, or SPA. HeLa LAMP1-GFP cells were infected with WT or *sifA* mutant strains of STM, STY, or SPA at MOI 50. Cells were fixed 8 h p.i., and microscopy was performed using a cell observer widefield system (Zeiss) with a 100x objective. At least 100 cells per strain were counted, and infected cells were scored for appearance of SIF (black bars) or absence of SIF (open bars). **C)** Induction of *sifA* expression by intracellular *Salmonella*. HeLa cells were infected with STM (green), STY (red), or SPA (blue) WT, each harboring dual fluorescence reporter p5633 for constitutive expression of DsRed, and P$_{sifA}$-controlled expression of sfGFP. Host cells were lysed 8, 16, or 24 h p.i. as indicated, and released bacteria were subjected to FC analyses as described for **Fig 1**. The proportion of intracellular *Salmonella* positive for P$_{sifA}$ induction is expressed as percentage of the entire population. The X-mean sfGFP fluorescence of the induced subpopulation was quantified. In **B)** and **C)** means and standard deviations of three biological replicates are shown. Statistical analysis was performed by One-way ANOVA and are indicated as follows: n.s., not significant; *, P< 0.05; **, P< 0.01; ***, P< 0.001.

presence of late endosomal/lysosomal membrane markers. We investigated the formation of SIF in SPA- or STY-infected host cells. Endosomal remodeling depends on function of the SPI2-T3SS, and effector protein SifA is essential for SIF formation and integrity of the SCV [23,24]. Infected cells were analyzed for formation of SIF and we compared intracellular phenotypes of WT and Δ*sifA* strains of the various serovars. In infected HeLa or RAW cells, we observed formation of LAMP1-GFP-positive membrane compartments that completely

enclosed individual bacteria (**Fig 4A**). The formation of SIF was observed in host cells infected with STM, STY or SPA WT strains, while STY Δ*sifA* (**Fig 4A**) or SPA Δ*sifA* strains did not induce SIF. In RAW264.7 macrophages, LAMP1-positive compartments were observed after phagocytosis of STM, STY, or SPA WT or SPI2-T3SS-deficient strains (**S2 Fig**). We did not address SIF induction in macrophages, since fixed cells had to be used and fixation results in fragmentation of tubular endosomal structures in phagocytes [21].

The frequency of SIF-positive cells infected with WT or Δ*sifA* strains was quantified in HeLa cells at 8 p.i. (**Fig 4B**). In line with prior observations, STM WT induced SIF in the majority of infected cells, while tubular endosomes were almost absent after infection by the STM Δ*sifA* strain (ca 4% positive cells). SIF formation in cells infected by STY or SPA was much less frequent and only about 10% were scored positive. Cells infected with STY or SPA Δ*sifA* strains very rarely showed tubular endosomal compartments.

We next attempted to correlate the frequency of SIF formation to the level of translocation of SifA as key SPI2-T3SS effector for SIF formation. While translocated SifA was detectable in host cells infected with STM WT matching prior observations [25], amounts of SifA detected in host cell infected by STY or SPA were too low to allow proper comparative quantification. Thus, we introduced a new dual fluorescence reporter with sfGFP expression controlled by $P_{sifA}$. Induction of $P_{sifA}$::sfGFP by intracellular STM, STY and SPA was quantified (**Fig 4C**). The frequency of $P_{sifA}$-induced STY and SPA was significantly lower than for STM at 8 h and 16 h p.i., and frequency reached similar levels at 24 h p.i. The sfGFP intensities of $P_{sifA}$-induced *Salmonella* were significantly lower for STY and SPA compared to STM at all time points investigated. The delayed and lower level of expression of genes encoding effector proteins also explain the reduced frequency of active manipulation of the host cell endosomal system by intracellular STY and SPA.

We further characterized the SCV using canonical markers of the endocytic pathway. The small GTPases Rab7A and Arl8A were transiently transfected in HeLa cells. To mark late endosomes and lysosomes in HeLa cells, we used fluid phase marker Dextran-Alexa647. The association of Rab7A and Arl8A with SCV and SIF was frequently observed in cells infected with STY or SPA (**S3 Fig**).

We conclude that SPI2-T3SS and effector SifA can mediate endosomal remodeling by STY and SPA. However, the frequency of this manipulation of host cell functions by STY and SPA is highly reduced compared to STM. A role of SifA in maintaining the integrity of SCV containing STY or SPA cannot be deduced from our experimental data. The molecular composition of endosomal compartments modified by STM, STY or SPA is similar.

## SPA and STY induce the formation of double-membrane SIF

We have recently unraveled the unique architecture of SIF induced by STM in epithelial cells and macrophages [9]. In particular, tubular compartments were delimited by vesicular single membranes, and at later time points after infection, SIF composed of double membranes were frequently observed. We analyzed if SIF induced by intracellular STY and SPA are comparable in architecture, or distinct to SIF observed in STM-infected cells. We performed correlative light and electron microscopy (CLEM) analyses in order to reveal the ultrastructural features of SIF networks identified by light microscopy (**Fig 5**). Although the frequency of STY- or SPA-infected cells with SIF was low, light microscopy allowed identification of these events for subsequent transmission electron microscopy (TEM) analyses. As for STM, SCV harboring STY or SPA were continuous and in close contact to the bacterial envelope. In cells with an extensive SIF network, characteristic double-membrane SIF were observed in HeLa cells infected by STM, as well as by SPA, or STY (**Figs 5,** S4, S5 **and** S6). SIF extending into the cell

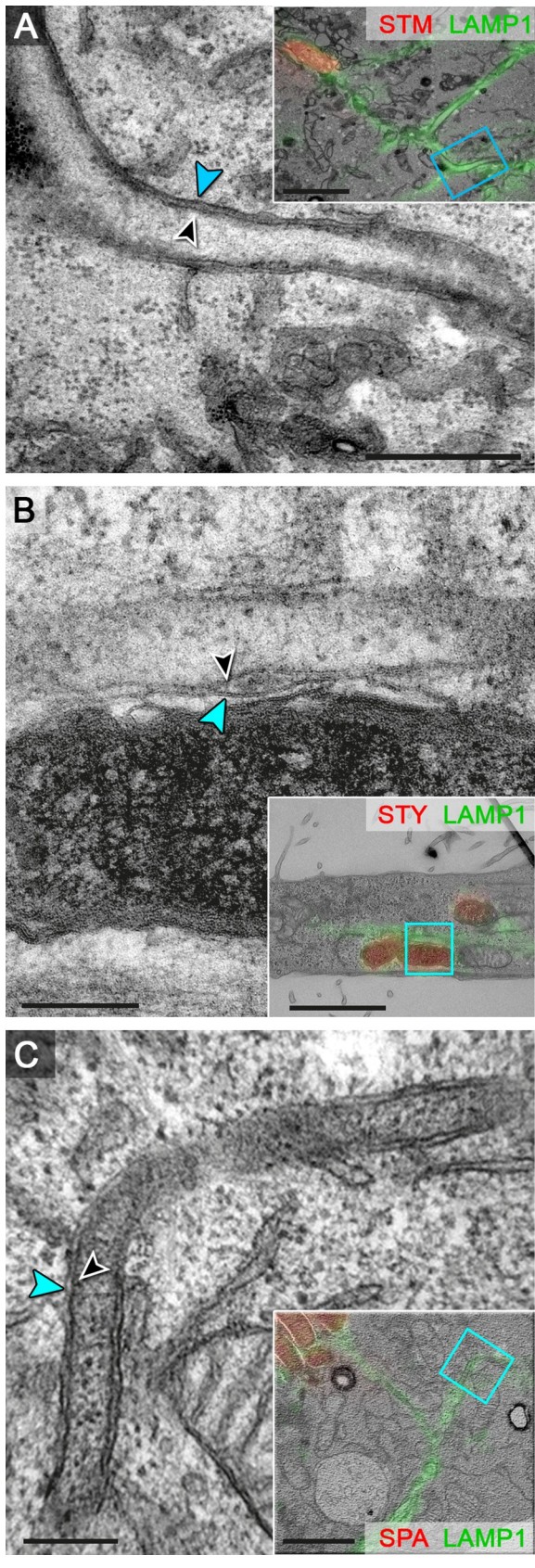

**Fig 5. Ultrastructure of endosomal compartments remodeled by intracellular STM, STY, or SPA.** HeLa cells expressing LAMP1-GFP (green) were infected with STM (**A**), STY (**B**), or SPA (**C**) WT, each expressing mCherry (red). Cells were fixed 7 h p.i. and processed for confocal fluorescence microscopy (FM) and transmission electron microscopy (TEM) of ultrathin sections. FM and TEM modalities were superimposed (inserts) for correlation, and details of the ultrastructure of tubular LAMP1-positive membrane compartments are shown at higher magnification. Double membrane SIF with inner (black arrowhead) and outer (blue arrowhead) membrane. Scale bars: 200 nm (details), 2 μm (inserts for overview).

periphery were connected to the SCV. The morphological characteristics of the tubular compartments appeared comparable for the three serovars, as well as the diameter of app. 160 nm.

We conclude that STY and SPA can induce tubular endosomal networks comparable to those characterized in STM-infected cells.

## SCV integrity and cytosolic access of intracellular STY and SPA

Maintenance of SCV integrity is important for virulence of STM, and mutant strains deficient in SPI2-T3SS effector protein SifA are more frequently released into the cytosol of host cells. Host cells respond to cytosolic STM with xenophagic clearance, or induction of pyroptotic cell death [12]. In epithelial cells, cytosolic hyper-replication of STM can lead to killing of host cells, or triggers expulsion of infected cells from polarized epithelial layers [26]. In contrast to STM, the importance of SCV integrity and potential contribution of SifA is not known for TS.

We recently used a dual fluorescence reporter plasmid for analyses of exposure of intracellular bacteria to host cell cytosol as result of impaired SCV integrity [27]. The sensor is based on the promoter of *uhpT*, a transporter for glucose-6-phosphate (G6P). As G6P is present in the cytosol of mammalian host cells, but is rapidly metabolized in bacterial cells, specific induction of this reporter occurs in STM that are in contact with host cell cytosol.

We tested the dual fluorescence cytosolic sensor in SPA and STY using *in vitro* growth conditions with various amounts of G6P (**Fig 6A**). Analyses by FC showed a G6P concentration-dependent increase in sfGFP fluorescence in STY similar to prior observations for STM [27], while no induction was observed for SPA. Analyses of the genome sequence of SPA indicated that the sensor-regulator system encoded by *uhpABC* is defective by pseudogene formation [28], and activation of P$_{uhpT}$ is not possible. Thus, the P$_{uhpT}$–based cytosolic sensor was not applied to analyses of SPA.

We compared cytosolic exposure of STM and STY in HeLa cells (**Fig 6B** and **6C**), or U937 macrophages (**Fig 6D** and **6E**). A rather large cytosolic-induced population was observed for STM, which increased over time. As a STM Δ*sifA* strain was reported to be deficient in maintaining the integrity of the SCV, we also determined cytosolic release of a STY Δ*sifA* strain. In HeLa cells infected with STM WT two populations were apparent, a sfGFP-positive population indicating cytosolic exposure, and one population without sfGFP fluorescence indicating segregation from host cell cytosol. At 16 h or 24 h p.i., STM Δ*sifA* was homogenously sfGFP-positive. For STY WT, only a small population was sfGFP-induced and this population decreases over time p.i. Only very low numbers of sfGFP-positive, cytosol-exposed STY Δ*sifA* were detected at any timepoint. In U937 cells, a sfGFP-positive population was only detected for STM Δ*sifA* at 16 or 24 h p.i. Cytosolic presence of *Salmonella* in phagocytes is known to trigger pyroptosis [29], and this response was also induced by STM Δ*sifA*. However, since FC was used to analyze the same number of infected cells for the various conditions, this likely revealed the subpopulation of pre-pyroptotic macrophages harboring cytosolic-induced STM Δ*sifA*. STM WT and STY WT and Δ*sifA* strain did not show sfGFP fluorescence signals, indicating very low numbers of intracellular *Salmonella* exposed to host cell cytosol.

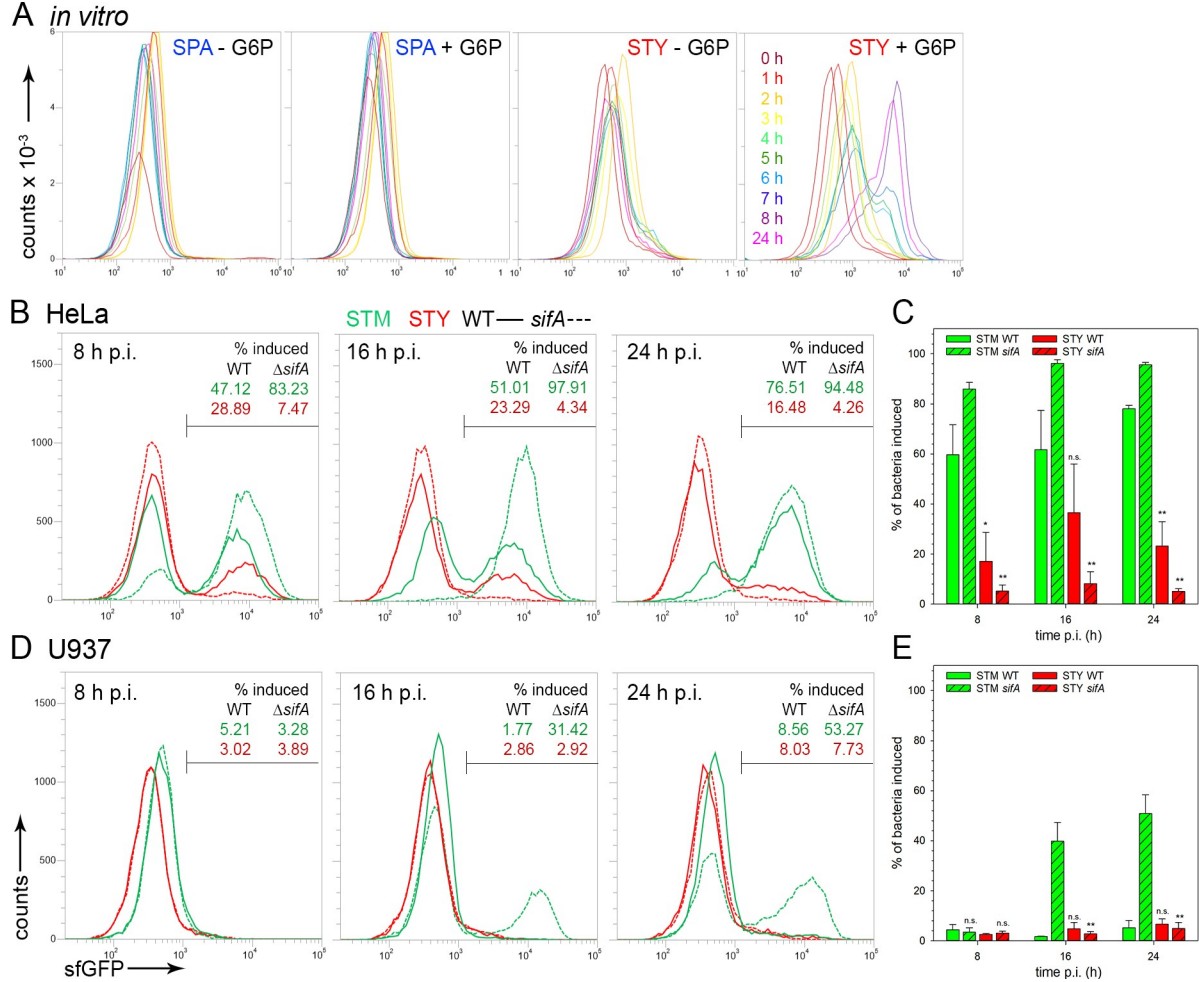

**Fig 6. SCV integrity and cytosolic release of STM and STY.** STM (green), SPA (blue), or STY (red), WT (solid lines) or Δ*sifA* (dashed lines) strains were used as indicated, each harboring plasmid p4889 for constitutive expression of DsRed, and $P_{uhpT}$–controlled expression of sfGFP as sensor for cytosolic exposure. **A)** Induction of $P_{uhpT}$::sfGFP by glucose-6-phosphate (G6P) in SPA and STY. SPA or STY WT harboring p4889 were grown in LB broth without or with addition of 0.2% G6P. Samples were collected at various time points of subculture as indicated by various colors. Bacteria constitutively expressing DsRed were analyzed by FC for levels of sfGFP expression. **B, D)** WT (solid lines) or *sifA* mutant (dashed lines) strains of STM (green) or STY (red), each harboring plasmid p4889 were used to infect HeLa cells (**B**), or U937 macrophages without IFN-γ-activation (**D**). At 8 h, 16 h, or 24 h p.i. as indicated, host cells were lysed in order to release bacteria. For FC, at least 10,000 bacteria-sized particles with DsRed fluorescence were gated and the GFP intensity was quantified. The percentage of bacterial cells with induction of $P_{uphT}$::sfGFP is indicated for WT and *sifA* mutant strains. The data sets shown in **B)** and **D)** are representative for three independent experiments with similar outcome. Cytosolic-induced *Salmonella* are expressed as percentage of entire intracellular *Salmonella*, means and standard deviations were determined for three independent experiments for infection of HeLa cells (**C**), or U937 cells (**E**). Statistical analysis was performed by One-way ANOVA and are indicated as follows: n.s., not significant; *, P< 0.05; **, P< 0.01; ***, P< 0.001.

We conclude that SCV harboring STY are less prone to damage, and less events of cytosolic access of STY occur. This can be explained by lower levels of proliferation of intracellular STY, and thus lower dependency on SifA function to maintain the integrity of the expanding SCV.

## The intracellular proliferation of SPI2-T3SS-deficient STY is reduced in HeLa cells

Prior work corroborated that function of SPI2-T3SS is required for intracellular proliferation of STM in various cell lines and in tissue of infected hosts [30]. The role of SPI2-T3SS for

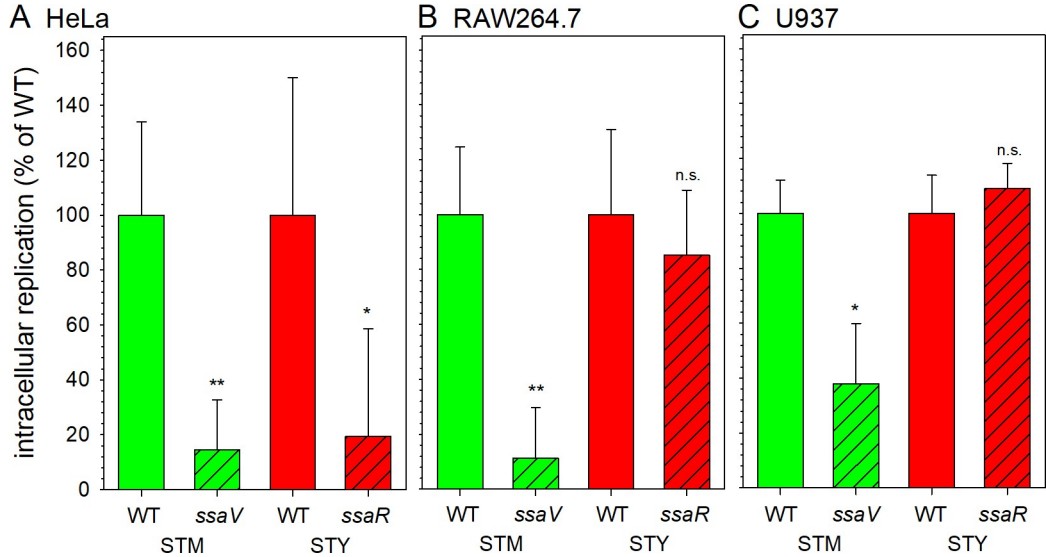

**Fig 7. SPI2-T3SS-deficient STY are reduced in intracellular replication in HeLa cells.** Hosts cells were infected with
WT (open bars) or SPI2-T3SS-deficient (hatched bars, *ssaV* and *ssaR* encode subunits of the SPI2-T3SS) strains of STM
(green) or STY (red) at MOI 1. Intracellular replication was determined by gentamicin protection assays comparing
intracellular CFU recovered 1 h p.i. and 24 h p.i. **A)** Intracellular replication in HeLa cells. STM was cultured overnight,
diluted 1:31 in fresh LB broth, subcultured 2.5 h, and used for infection. STY was incubated in 3 ml LB broth for 8 h,
reinoculated 1:100 in 10 ml fresh LB broth and grown for 16 h under microaerophilic conditions before used as inoculum
in infection **B, C)** Bacterial strains were grown overnight in LB and aliquots were used to infect macrophages. **B)**
Intracellular replication of STM and STY in RAW264.7 macrophages. **C)** Intracellular replication of STM and STY in
U937 cells. Assays were performed in three biological replicates and intracellular proliferation is expressed as percentage of
WT. Student´s *t*-test was used for statistical analysis and significance is indicated as follows: n.s., not significant; *,
P < 0.05; **, P< 0.01.

intracellular proliferation of TS is still open. This lack of information, and functionally of the
SPI2-T3SS demonstrated here prompted us to test for STM, STY and SPA intracellular prolif-
eration as function of SPI2-T3SS by gentamicin protection assays (**Fig 7**). Intracellular prolif-
eration of SPI2-T3SS-deficient STM Δ*ssaV* in HeLa was reduced about 10-fold. SPI2-T3SS-
deficient STY Δ*ssaR* was similar with about 8-fold attenuation compared to WT. While SPI2-
T3SS-deficient STM were highly and moderately attenuated in proliferation in RAW264.7 and
U937 macrophages, respectively, no attenuation was observed for STY *ssaR*.

The data shown in **Figs 1** and **4** indicate that only a small fraction of intracellular STY or
SPA deploys the SPI2-T3SS. Thus, the intracellular proliferation of STY may be low overall,
and the defect in proliferation of a SPI2-T3SS-deficient strain may be masked by a large num-
ber of intracellular STY that survive but fail to proliferate. To address intracellular proliferation
on the levels of single host cells, we deployed FC and determined mCherry-mediated fluores-
cence as proxy for bacterial load. We recently compared properties of various FP and identi-
fied fast-maturing, constitutively expressed mCherry as reliable indicator for proliferation of
intracellular bacteria [31].

Analyses of single host cells (**Figs 8** and **S7**) confirmed the SPI2-T3SS-dependent intra-
cellular proliferation of STM in HeLa, RAW264.7 and U937, and of STY in HeLa. The
X-means for RAW264.7 or U937 harboring STY also increased from 1 h p.i. to 24 h p.i. This
increase was rather small and independent from function of the SPI2-T3SS. Interestingly,
STM also showed SPI2-T3SS-independent increase of bacterial mCherry fluorescence in
HeLa, RAW264.7, and U937 cells, but the increase was always reduced compared to prolifera-
tion of STM WT.

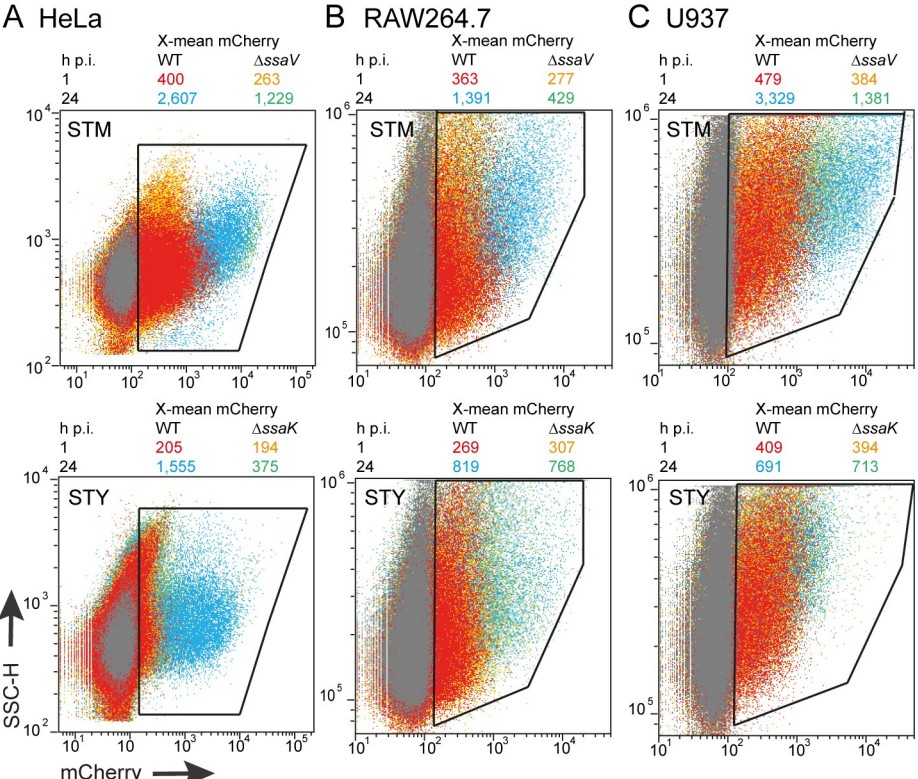

**Fig 8. Intracellular proliferation of STM and STY in HeLa cells, RAW264.7 macrophages, or U937 macrophages determined flow cytometry.** For infection of HeLa cells (**A**) STM was cultured overnight in 3 ml LB broth and reinoculated 1:31 in fresh LB broth for 3.5 h. STY was incubated in 3 ml LB broth for 8 h and reinoculated 1:100 in 10 ml fresh LB broth for 16 h under microaerophilic conditions. For infection of RAW264.7 macrophages without IFN-γ-activation (**B**), or U937 macrophages without IFN-γ-activation (**C**), overnight cultures were used. At 1 h or 24 h p.i. as indicated, cells were washed, fixed and detached. The cell populations were analyzed by flow cytometry and gating was set to cells harboring intracellular *Salmonella* as indicated by polygons. At least 10,000 *Salmonella*-infected host cells were scored. Overlays of dot plots show WT *Salmonella* at 1 h or 24 h p.i. in red and blue, respectively, and Δ*ssaV*/Δ*ssaK Salmonella* at 1 h or 24 h p.i. in orange and green, respectively. The X-mean values for mCherry fluorescence intensities of infected cells are shown in same colors above each overlay. Individual dot plots are shown in **S7 Fig**. Assays were performed in three biological replicates, one representative replicate is shown.

Taken together, these results support a role of SPI2-T3SS in intracellular proliferation of STY in HeLa cells, and redundancy of SPI2-T3SS function for proliferation in the phagocytic cell lines investigated.

## Biosynthetic capability of intracellular STM, SPA and STY

Finally, we set out to determine the proportion of intracellular STM, SPA or STY that remain capable of protein biosynthesis upon application of an external stimulus. We anticipated that dead bacteria are unable to respond to an external stimulus, and those that have entered a persister state are reduced in protein biosynthesis. Although intracellular persisters of STM are able to activate virulence gene expression [32] and stress response system [33], the overall biosynthetic capacity decreased during intracellular presence [33]. As experimental system, we introduced in STM, STY and SPA a dual fluorescence plasmid for constitutive expression of DsRed, and expression of sfGFP under control of the promoter of *tetA*. We recently demonstrated that P*tetA* can be efficiently activated in intracellular *Salmonella* by external addition of the inducer anhydrotetracycline (AHT) [34]. Under culture conditions in medium, sfGFP

synthesis was induced by addition of AHT in STM, SPA or STY to comparable levels (**Fig 9A**). For SPA induced by various concentrations of AHT, a small subpopulation was observed with lower levels of sfGFP fluorescence. After infection of various host cell types by STM, SPA or STY, inducer AHT was added to infected cells 1.5 h prior to the end of the infection period and lysis of host cells (**Fig 9B**). In HeLa cells 8 h p.i., around 80% of the bacteria showed induction, and the intensities for STY cells were higher than for STM and SPA (**Fig 9C** and **9D**). The proportion of induced STM remained high at 16 h and 24 h p.i., while the percentage of AHT-induced STY and SPA declined at 16 h and 24 h p.i., as well as sfGFP intensities of induced *Salmonella*.

In human U937 macrophages (**Fig 9E** and **9F**), a similar percentage of AHT-inducible bacteria was determined 8 h p.i. for all serovars. At 16 h p.i., the amounts of AHT-inducible STY and SPA dropped to 62.4% and 58.2%, respectively, while STM remained at more than 91.4% AHT-inducible bacteria. At 24 h p.i. we observed an increase in the percentage of AHT-inducible STY and SPA to 81.7% and 71.0%, respectively. The proportion of response of STM remained constantly high.

In murine RAW264.7 macrophages (**Fig 9G** and **9H**), we observed a high percentage of AHT-inducible STM (62.8 ± 0.5%) and STY (59.5 ± 13.2%) at 8 h p.i., while only 4.9 ± 5.7% of SPA were induced. At 24 h p.i., a decrease in AHT-induced STY was observed with 29 ± 3.5% sfGFP-positive bacteria, and 53 ± 27.3% AHT-induced STM were determined. It should be noted that sfGFP fluorescence levels decreased from 8 h to 24 h p.i. for AHT-induced STY and SPA (**Fig 9G**). For SPA, the AHT-inducible population remained very low at 24 h p.i. (± 3.9 ± 0.9%).

To test if biosynthetic activity of intracellular *Salmonella* is affected by functionality of the SPI2-T3SS, we compared sfGFP expression upon external induction for STM WT and Δ*ssaV* strains (**S8 Fig**). We observed that response to AHT was rather similar for STM WT and Δ*ssaV* in U937 or RAW264.7 cells various at various time points after infection. In HeLa cells, the percentage of induced STM was similar for WT and Δ*ssaV* at 3 h and 8 h p.i. At later time points p.i., a higher proportion of induced WT was observed, while proportion of induced STM Δ*ssaV* remained at 45.5 to 49.5%. We conclude that the acquisition of host cell nutrients by STM WT deploying SPI2-T3SS increases the biosynthetic capacity in HeLa cells, while this effect was not detected in the more restrictive environment of phagocytic cells.

The analyzed serovars are rather distinct in their ability to respond to intracellular environments. STM appeared most robust and biosynthetic active populations of similar size were detected in all host cell types and any time point. STY showed highest percentage of biosynthetic active bacteria at early time points in all cell lines analyzed, and the proportion of AHT-inducible cells decreased in all host cell types at 16 and 24 h p.i. The level of sfGFP expression of the proportion of AHT-inducible SPA was already reduced at early time points and further decreased over time, indicating a continuous loss of biosynthetic capacity over time. The effect was most pronounced in RAW264.7 cells. We conclude that STY and SPA have a lower capacity than STM to maintain a biosynthetic active state over prolonged presence in host cells.

## Discussion

Our study investigated the intracellular lifestyle of STY and SPA as clinically most relevant serovars of TS. We demonstrated that SPI2-T3SS is functional in translocating effector proteins that induce endosomal remodeling, and formation and maintenance of SCV. The SPI2-T3SS-dependent remodeling of the endosomal system by STY and SPA was reminiscent of the phenotypes observed in STM-infected cells. By this, we provide formal proof of the function of the SPI2-T3SS in human-adapted TS serovars. As major difference in intracellular phenotypes of

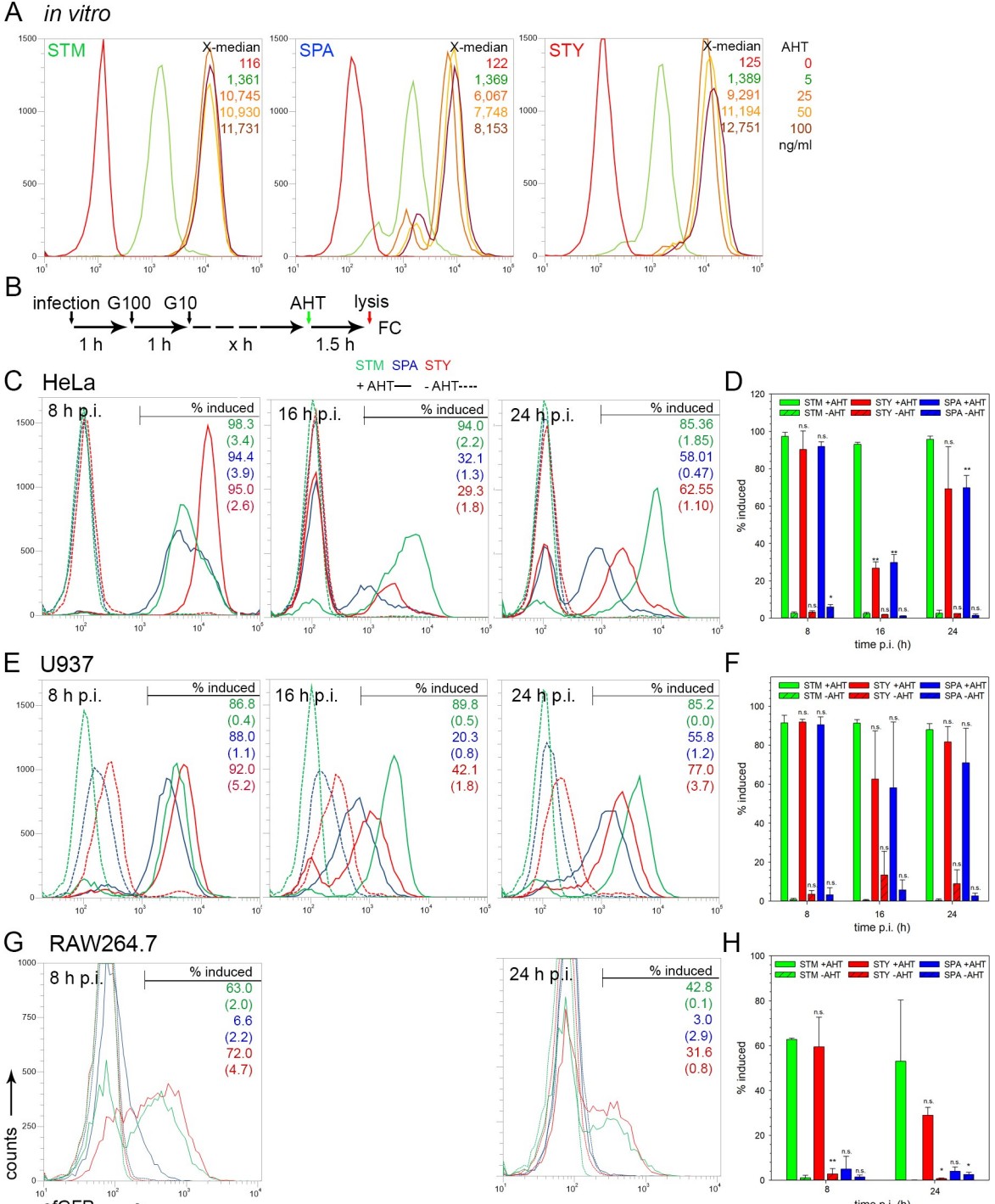

**Fig 9. Biosynthetic capability of intracellular STM, SPA and STY. A**) STM (green), SPA (blue), and STY (red) WT strains, each harboring plasmid p4928 for constitutive expression of RFP, and sfGFP under control of the *tetA* promoter were subcultured in LB broth with aeration by rotation at 60 rpm in a roller drum. Various amounts of AHT as indicated were added after 1.5 h of culture. Culture was continued for 2 h, bacterial cells were fixed and the sfGFP fluorescence intensity was determined by FC for at least 50,000 DsRed-positive bacteria per condition. The X-median values of sfGFP fluorescence are shown for a representative experiment. **B**) STM, SPA, or STY WT strains harboring p4928 were used to infect HeLa cells (**C**), U937 cells without IFN-γ-activation (**E**), or RAW264.7 cells without IFN-γ-activation (**G**). After infection for 1 h, medium was exchanged against medium containing 100 μg x ml$^{-1}$ gentamicin (G100) for 1 h, followed by medium containing 10 μg x ml$^{-1}$ gentamicin for the rest of the experiment. For induction of sfGFP expression, 50 ng x ml$^{-1}$ AHT was added to infected cells 1.5 h prior lysis of host cells. At the time points p.i. indicated, host cells were lysed in order to release

intracellular bacteria. For FC, at least 10,000 bacteria-sized particles were gated based on DsRed fluorescence, and the sfGFP intensity was quantified. The percentage of sfGFP-positive bacteria is indicated and values in brackets show the non-induced controls. The data sets shown are representative for three independent experiments with similar outcome. AHT-induced *Salmonella* are expressed as percentage of entire intracellular *Salmonella*, means and standard deviations were determined for three independent experiments for infection of HeLa cells (**D**), U937 macrophages (**F**), or RAW264.7 macrophages (**H**). Statistical analysis was performed by One-way ANOVA and are indicated as follows: n.s., not significant; *, $P < 0.05$; **, $P < 0.01$; ***, $P < 0.001$.

SPA and STY compared to STM, we observed the delayed activation and overall lower frequency of intracellular *Salmonella* showing activity of SPI2-T3SS and related phenotypes. Furthermore, the population of intracellular STY exhibits larger heterogeneity. In the following, these aspects are discussed in more detail.

## Low intracellular activity of TS

Compared to STM, intracellular STY and SPA formed smaller populations of bacteria with SPI2-T3SS function and more general, biosynthetic activity in response to artificial induction. SPI2-T3SS function mediated proliferation in HeLa but not in the more restrictive environment of macrophages. These observations indicate that STY and SPA are either more efficiently killed by host cells, or more frequently develop persister state. Persister bacteria are key contributors to persistent as well as recurring infections [35], and both phenomena are characteristic to infectious diseases caused by TS [4]. Future studies have to reveal the role of persister formation of intracellular STY and SPA and the contribution of persister bacteria to pathogenesis of TS infections.

## Role of the SPI2-T3SS in TS

We overserved promoters of SPI2-T3SS and effector genes were induced in intracellular STM, STY and SPA in various host cells. In contrast, translocation of SPI2-T3SS effector proteins occurred in a lower number of host cells infected by STY or SPA, and amounts of effector proteins were highly reduced. This may indicate that despite receiving the proper intracellular stimuli for activation of SsrAB-regulated genes, the subsequent syntheses of T3SS subunits, assembly or a functional T3SS, and/or translocation of effector proteins is less efficient in STY and SPA. The reduced intracellular proliferation of STY and SPA might be consequence of this difference.

We previously demonstrated that SPI2-T3SS-mediated SIF formation augments nutritional supply of STM in the SCV, and also leads to reduction of antimicrobial factors acting on bacteria in the SCV [36,37]. Since SIF formation is less frequent in host cells infected by STY or SPA compared to STM, this may indicate a restricted nutritional supply, leading to limited intracellular proliferation. As STY and SPA are auxotrophic for cysteine and tryptophan [38] the demand for external supply with amino acids is higher than for prototrophic STM. Furthermore, exposure to effectors of host cell defense mechanisms may be increased for STY or SPA in SCV without SIF connection. Further single cell analyses can address this potential correlation between SIF formation and proliferation for intracellular TS.

Our work confirms prior reports that SPI2-T3SS is not required for net survival and proliferation of STY in infection models with human macrophages [6]. Our observation that only a small percentage of bacteria synthesize SPI2-T3SS and deploy its function would also allow alternative explanations. In ensemble-based analyses such as the gentamicin protection assay, ongoing proliferation in a small number of infected host cells may by masked by the majority of infected cells that kill the pathogen or restrict its proliferation. Even if the number of permissive infected cells is low, these may play an important role for progression of a systemic

infection [39]. The application of single cell FC analyses will allow future in-depth analyses of the fate of TS in distinct subpopulations of host cells.

## Role of SPI2-T3SS effector proteins in TS

The SPI2-T3SS in STM translocates a complex set of more than 30 effector proteins into infected host cells. These effector proteins form subsets that act on the endosomal system, affect innate immune signaling, formation of adaptive immunity, interfere with ubiquitination, or have functions still unknown [5]. The repertoire of effector proteins in STY and SPA is severely restricted, with intact genes only for 11 or 8 effectors proteins in STY and SPA, respectively [5]. This observation may reflect the adaptation to a narrow host range accompanied by loss of redundant effectors. Of note, the subset of effector proteins involved in endosomal remodeling (SseF, SseG, SifA, PipB2, SteA) is present. We showed that SifA is required for induction of SIF formation in STY and SPA similar to STM. However, our analyses did not support a contribution of SIF in maintaining the SCV integrity in cells infected by STY. Among other sequence variations, the altered sequence of the C-terminal membrane anchoring hexapeptide [40] may affect function of SifA in STY.

Also present in STM, STY and SPA is SteD, an effector protein modulating adaptive immunity [41]. The activation of CD4+ T cells is very important for the elimination of *Salmonella* in mice and human. In STM, SteD stimulates the ubiquitination of MHCII, leading to its degradation and therefore prevention of antigen presentation by MHCII [41]. SteD is present in STY and SPA, but shows subtle differences in amino acid sequence. Thus, it is not certain whether SteD exhibits the same function in STY or SPA as in STM. Because interference with host adaptive immunity is of central importance of persistent infections as caused by TS, the function of SteD in STY and SPA deserves further investigation.

It was reported that effector protein SteE of STM directs macrophage polarization towards an anti-inflammatory M2 state [32]. However, STY and SPA lack SteE and are unlikely to manipulate the activation state of macrophages by this mechanism. Uptake by macrophages may be fatal for individual STY or SPA during infection, and phagocytic cells with other properties, such as dendritic cells, could form important vehicles of systemic distribution of TS. Again, host specificity and low numbers of STY- or SPA-infected cells in human blood are obstacles to more detailed analyses.

Additional effector proteins that are specific to STY and SPA may be present and mediate serovar-specific properties. One example is StoD, an E3/E4 ubiquitin ligase that is translocated by the SPI1-T3SS [42]. A systematic screen by bioinformatics tools and experimental validation may identify TS-specific effector proteins of SPI2-T3SS.

## Need of suitable infection models for TS

We observed distinct intracellular phenotypes in the various cell lines and primary cells used in this study. However, the specific virulence properties of TS and the contribution of SPI2-T3SS require analyses in improved infection models. Of specific interest will be organoids of human origin that are capable to simulate tissues that are infected by STY or SPA. In the human body, STY persistently colonizes gallbladder and bone marrow. *Salmonella* is resistant to high concentrations of bile and bile also has an influence on the invasion of epithelial cells by *Salmonella* [43]. Bile induces persister cell formation of STY and an associated tolerance to antibiotics [44]. It is also known that STY forms biofilm in the gallbladder and on gallstones. This could lead to constant inflammation of gallbladder tissue, and during persistent infection to development of gallbladder cancer [45]. The formation of biofilm enables STY to adapt a carrier state in the host, which occurs in about 3–5% of the infected people [43]. For

example, a recently introduced organoid model of human gallbladder epithelium may provide new options to study TS in a setting of a persistent infection [46]. If SPI2-T3SS effector functions beyond intracellular survival and proliferation should be analyzed, even more complex experimental systems with human immune cells are required. As alternative, a humanized mouse model based on transplantation of human immune cells may be considered [7]. A genome-wide screen for genes required for survival in STY in humanized mice did not reveal contributions of SPI2-T3SS or effector protein. Since the initial screen was performed using an infection period of 24 h, identification of STY factors affecting formation of adaptive immune responses were not identified. Future analyses in the humanized mouse model under conditions allowing persistent infection will be of interest, but the high demands of the complex model may compromise more frequent applications.

## Concluding remarks

This study demonstrated that the functionality of SPI2-T3SS in STY and SPA, and future studies with improved infection models have to reveal the contribution of this virulence factor to pathogenesis of infectious diseases by TS. We applied a set of single cell approaches to study the intracellular adaptation strategies of STM as NTS serovar in comparison to TS serovars STY and SPA. These analytic tools enabled a detailed view on the specific intracellular activities of NTS and TS, and will enable future in-depth characterization of bacterial heterogeneity and adaptation strategies.

## Materials and methods

### Bacterial strains and culture conditions

For this study *Salmonella enterica* serovar Typhimurium (STM) NCTC12023, *S. enterica* serovar Typhi (STY), and *S. enterica* serovar Paratyphi A (SPA) were used as wild-type (WT) strains. All mutant strains are isogenic to the respective WT and **Table 1** shows characteristics

**Table 1. Bacterial strains used in this study.**

| Designation | relevant characteristics | reference |
|---|---|---|
| *S. enterica* serovar Typhimurium | | |
| NCTC12023 | wild type | Lab collection |
| P2D6 | *ssaV*::mTn5 | [30] |
| MvP503 | Δ*sifA*::FRT | [20] |
| *S. enterica* serovar Typhi | | |
| 120130191 | wild type | clinical isolate, SalHostTrop consortium |
| STY101 | Δ*ssaR*::aph | this study |
| STY118 | Δ*ssaR*::FRT | this study |
| STY110 | Δ*sifA*::aph | this study |
| STY123 | Δ*sifA*::FRT | this study |
| STY132 | Δ*ssrB*::aph | this study |
| STY134 | Δ*ssaK*::aph | this study |
| STY137 | Δ*ssaK*::FRT | this study |
| *S. enterica* serovar Paratyphi A | | |
| 45157 | wild type | clinical isolate, SalHostTrop consortium |
| SPA118 | Δ*ssaR*::FRT | this study |
| SPA110 | Δ*sifA*::aph | this study |
| SPA132 | Δ*ssrB*::aph | this study |

of strains used in this study. STM, STY and SPA strains were routinely grown on Luria-Bertani broth (LB) agar or in LB broth containing 50 μg x ml$^{-1}$ carbenicillin for maintenance of plasmids at 37˚C using a roller drum. As synthetic media, PCN (Phosphate, Carbon, Nitrogen) was used as described before [47] with the indicated pH concentrations of inorganic phosphate. For culture of STY and SPA, PCN media were supplemented with 20 μg x ml$^{-1}$ of each cysteine and tryptophan.

## Generation of bacterial strains

Mutant strains harboring deletions in various virulence genes were generated by λ Red recombineering for insertion of the kanamycin resistance cassette amplified from template plasmid pKD13 basically as described before [48,49] using oligonucleotides listed in **S1 Table**. The proper insertion was confirmed by colony PCR. If required, the *aph* cassette was removed by introduction of pE-FLP and FLP-mediated recombination. Plasmid p5633 as reporter for expression of *sifA* was generated by Gibson assembly for exchange of P$_{uhpT}$ in p4889 by a 300 bp PCR fragment containing P$_{sifA}$, basically as described before [27,50]. Plasmids used in this study are listed in **Table 2** and were introduced into various WT and mutant strains by electroporation.

## Host cell culture and infection

The murine macrophage cell line RAW264.7 (American Type Culture Collection, ATCC no. TIB-71) was cultured in high-glucose (4.5 g x ml$^{-1}$) Dulbecco's modified Eagle's medium (DMEM) containing 4 mM stable glutamine (Merck) and supplemented with 6% inactivated fetal calf serum (iFCS, Sigma). The human macrophage-like cell line U937 (ATCC no. CRL-1593.2) was cultured in RPMI-1640 medium (Merck) supplemented with 10% iFCS. Human primary macrophages were generated from monocytes isolated from buffy coat as previously described [50], and cultured in RPMI-1640 medium supplemented with 20% iFCS. Under these conditions, predominantly M1-polarized macrophages were obtained. The non-polarized epithelial cell line HeLa (ATCC no. CCL-2) was cultured in high-glucose DMEM containing 4 mM stable glutamine, sodium pyruvate and supplemented with 10% iFCS. Stably

**Table 2. Plasmids used in this study.**

| Designation | relevant characteristics | reference |
| --- | --- | --- |
| pFPV-mCherry | const. mCherry | [52] |
| pWRG167 | P$_{EM7}$::sfGFP in pWRG81 | [53] |
| pMW211 | const. DsRed T3_S4T | [54] |
| pWRG730 | Red recombinase expression | [55] |
| pE-FLP | FLP recombinase expression | [56] |
| pGL-Rab7 wt | Rab7a::GFP | [57] |
| p2095 | P$_{sseA}$ *sscB sseF*::M45 | [18] |
| p2129 | P$_{sseJ}$ *sseJ*::M45 | [18] |
| p2621 | P$_{pipB2}$ *pipB2*::M45 | [19] |
| p3301 | P$_{sseL}$ *sseL*::HA | Lab collection |
| p3774 | const. RFP | Lab collection |
| p3776 | P$_{EM7}$::RFP P$_{ssaG}$::sfGFP | [27] |
| p4514 | Arl8A::eGFP | this study |
| p4889 | P$_{EM7}$::DsRed P$_{uhpT}$::sfGFP | [27] |
| p4928 | P$_{EM7}$::RFP *tetR* P$_{tetA}$::sfGFP | [34] |
| p5633 | P$_{EM7}$::DsRed P$_{sifA}$::sfGFP | this study |

transfected HeLa cell lines expressing LAMP1-GFP were cultured under the same conditions. All cells were cultured at 37˚C in an atmosphere containing 5% $CO_2$ and absolute humidity.

## Gentamicin protection assay

The assay was performed as described before [20]. Briefly, RAW264.7 cells were seeded 24 h or 48 h prior infection in a surface-treated 24-well plate (TPP) to reach confluency (~4 x $10^5$ cells per well) on the day of infection. U937 cells were seeded 72 h prior infection in surface-treated 24-well plates (TPP) to reach confluency (~4 x $10^5$ cells per well) on the day of infection and were treated with 50 ng x $\mu l^{-1}$ phorbol 12-myristate 13-acetate (PMA) for differentiation and cell attachment. HeLa cells were seeded 48 h prior infection in surface-treated 24-well plates (TPP) to reach confluency (~2 x $10^5$ cells per well) on the day of infection. For infection of RAW264.7 and U937 cells, bacteria were grown overnight (~ 20 h) aerobically in LB medium. For infection of HeLa cells, fresh LB medium was inoculated 1:31 with overnight cultures of STM and incubated for 3.5 h with agitation. For subculture of STY and SPA, 10 ml fresh LB medium was inoculated 1:100 with aerobic over day cultures and grown static under microaerophilic conditions for 16 h [51]. Then the bacteria were adjusted to an $OD_{600}$ of 0.2 in PBS and further diluted in DMEM (RAW and HeLa cells) or RPMI medium (U937 cells) for infection of cells at MOI of 1. Bacteria were centrifuged onto the cells for 5 min at 500 x g, and the infection was allowed to proceed for 25 min. After three washing steps with PBS, medium containing 100 µg x $ml^{-1}$ gentamicin was added for 1 h to kill extracellular bacteria. Afterwards the cells were incubated with medium containing 10 µg x $ml^{-1}$ gentamicin for the ongoing experiment. Cells were washed three times with PBS and lysed using 0.1% Triton X-100 at 2 h and 24 h post infection (p.i.). Colony forming units (CFU) were determined by plating serial dilutions of lysates and inoculum on Mueller-Hinton II agar and incubated overnight at 37˚C. The percentage of phagocytosed bacteria as well as the replication rate was calculated.

## Infection experiments for microscopy

HeLa cells stably transfected and expressing LAMP1-GFP were seeded in 24-well plates (TPP) on coverslips. The cells were grown to 80% confluency (~ 1.8 x $10^5$) on the day of infection. The cells were infected with STM, STY and SPA strains as described above with aerobic 3.5 h (for STM) and microaerobic, static 16 h (for STY, SPA) bacterial subcultures. For detection SPI2-T3SS effector protein translocation, MOI of 75 was used for infection and cells were analyzed 9 h p.i. For visualization of bacteria harboring reporter plasmids, MOI 50 was used for STM and SPA, and MOI 75 was used for STY. Afterwards the cells were washed three times with PBS and fixed with 3% paraformaldehyde (PFA) in PBS.

## Transfection

HeLa cells were cultured in 8-well dishes (ibidi) for one day. One µg of plasmid DNA of various transfection vectors was diluted in 25 µl DMEM without iFCS and mixed with 1 µl FUGENE reagent (ratio of 1:2 for DNA to FUGENE). The transfection mix was incubated for 10 min at room temperature (RT) and added to the cells in DMEM with 10% iFCS for at least 18 h. Before infection, the cells were treated with fresh medium.

## Pulse-chase with fluid phase markers

The fluid phase marker AlexaFluor 647-conjugated dextran (dextran-A647), molecular weight 10,000 (Molecular Probes) was used for tracing the endocytic pathway. HeLa cells were

incubated with 100 μg x ml$^{-1}$ dextran-A647 1 h p.i. until fixation of the cells. Subsequently, cells were washed and prepared for microscopy.

## Immuno-staining and fluorescence microscopy

Cells fixed with 3% PFA were washed three times with PBS and incubated in blocking solution (2% goat serum, 2% BSA and 0.1% saponin in PBS) for 30 min. Next, cells were stained for 1 h at RT with primary antibodies against *Salmonella* O-Ag of STM, STY and SPA (1:500), anti-HA (1:500) or anti-M45 (1:10). Accordingly, secondary antibodies were selected and samples were incubated for 1 h. Antisera and antibodies used in this study are listed in **S2 Table**. Coverslips were mounted with Fluoroprep (Biomerieux) and sealed with Entellan (Merck). The microscopy was performed by confocal laser-scanning microscopy (CLSM) on a Leica SP5 using the 100x objective (HCX PL APO CS 100 x, NA 1.4–0.7) and the polychroic mirror TD 488/543/633 for the three channels GFP, TMR/Alexa568, Cy5 (Leica, Wetzlar, Germany). For image processing, the LAS-AF software (Leica, Wetzlar, Germany) was used.

For analyses of effector translocation, HeLa cells LAMP1-GFP cells were infected and prepared for imaging as described before. Visualization was performed with Zeiss Cell Observer Spinning Disk (20% Laser, 100 ms exposure) with Plan-Apochromat 40x oil objective (NA 1.4). Sum projections of images were analyzed by FIJI by defining ROIs of infected cells and measuring area, mean grey value and integrated density. Corrected total cell fluorescence was calculated by subtracting area X-mean fluorescence of background ROI from integrated density of infected cell ROI.

## Correlative light an electron microscopy (CLEM)

CLEM of HeLa LAMP1-GFP cells infected by STM, SPA, or STY was performed as previously described [9]. Briefly, HeLa LAMP1-GFP cells were grown on MatTek dishes with gridded coverslips and infected with the respective Salmonella strains at MOI 75. Cells were fixed with 2% paraformaldehyde (PFA) and 0.2% glutaraldehyde (GA) in 200 mM HEPES for 30 min prior to fluorescence microscopy. After rinsing the cells thrice with 200 mM HEPES buffer, free aldehydes were blocked by incubation with 50 mM glycine in buffer for 15 min, followed by rinses in buffer. CLSM was performed and ROIs were chosen. Cells were subsequently fixed with 2.5% glutaraldehyde and 5 mM CaCl$_2$ in 200 mM HEPES for 1 h for TEM. Further steps including post-fixation, dehydration, sectioning and imaging by TEM were perfromed as previously described [9].

## Flow cytometry analyses

Cells were seeded in 12-well plates (TPP) 48 h (HeLa and RAW264.7), or 72 h (U937) prior infection to reach confluency on the day of infection (HeLa cells: 4 x 10$^5$ cells/well, RAW264.7: 8 x 10$^5$ cells/well, U937: 7 x 10$^5$ cells/well). U937 cells were treated with 50 ng x μl$^{-1}$ PMA. RAW264.7, HeLa or U937 cells were infected with STM, STY and SPA WT or ΔssrB and ΔsifA strains at MOI of 30 as described before. The bacterial strains harbored plasmids constitutively expressed RFP (p3776, p4928) or DsRed (p4889, p5633), and expressed sfGFP after activation of respective promotors. To determine metabolic activity, cells infected with *Salmonella* harboring plasmid p4928 were induced 2 h before fixation by addition of AHT to 50 ng x ml$^{-1}$. At 8, 16, or 24 h p.i., cells were washed with PBS, detached from the culture plates, lysed with 0.1% Triton X-100, fixed with 3% PFA and analyzed by FC using an Attune NxT cytometer (Life Technologies, ThermoFisher). At least 10,000 RFP- or DsRed-positive cells were measured and the cells expressing red and green fluorescence were analyzed.

## Flow cytometry analyses of whole host cells

Cells were seeded in 6-well plates (TPP) 48 h (HeLa and RAW264.7), or 72 h (U937) prior infection to reach confluency on the day of infection (HeLa cells: 8 x $10^5$ cells/well, RAW264.7: 1.6 x $10^6$ cells/well, U937: 1.4 x $10^6$ cells/well). U937 cells were treated with 50 ng x $\mu l^{-1}$ PMA. RAW264.7, HeLa or U937 cells were infected with STM, STY and SPA WT or SPI2-T3SS-deficient strains ($\Delta ssaV$, $\Delta ssaK$, or $\Delta ssaR$) at MOI 10 as described before. The bacterial strains harbored plasmids constitutively expressing mCherry. At 1 h and 24 h p.i., cells were washed with PBS, detached form the culture plates, fixed with 3% PFA and analyzed by FC using an Attune NxT cytometer. At least 10,000 mCherry-positive cells were measured and intensities of mCherry fluorescence signals per host cell were quantified.

## Data analyses

Statistical analyses were performed using SigmaPlot 13 by One-way ANOVA with Bonferroni t-test.

## Supporting information

**S1 Fig. Induction of $P_{ssaG}$ by intracellular STM, STY or SPA.** HeLa cells were infected at MOI 75 with STM (**A**), STY (**B**), or SPA (**C**), WT or $\Delta ssrB$ strains all harboring plasmid p3776 for constitutive expression of RFP (red), and sfGFP (green) under control of the *ssaG* promoter. To label the endosomal system, cells were pulsed with Dextran-Alexa647 (Dextran-A647, blue) from 1 h p.i. until fixation. Cells were fixed with PFA 8 h p.i. and microscopy was performed by CLSM on a Leica SP5 using the 100x objective. Scale bars, 10 μm.
(TIF)

**S2 Fig. Formation of SCV in RAW264.7 macrophages infected with STM, STY, or SPA cells compared to STM.** RAW264.7 cells constitutively expressing LAMP1-GFP, without IFN-γ-activation, were infected by WT or SPI2-T3SS-deficient strains ($\Delta ssaV$, $\Delta ssaK$, or $\Delta ssaV$) of STM, STY, or SPA as indicated, each constitutively expressing mCherry. Cells were fixed 8 h p.i. and subjected to CLSM on a Leica SP5. 3D reconstructions of Z stacks are shown. Scale bars, 5 μm.
(TIF)

**S3 Fig. Vesicular compartments harboring STY or SPA are positive for canonical SCV markers Arl8A and Rab7.** HeLa cells were transiently transfected with plasmids for the expression of Arl8A-eGFP or Rab7A-eGFP as indicated, and infected with STY (**A**) or SPA (**B**) at MOI 75 (for STY) or 50 (for SPA). Infected cells were pulsed-chased with Dextran-Alexa647 from 1 h p.i. until fixation. Microscopy was performed by CLSM on a Leica SP5 using a 100 x objective. Scale bars, 10 μm (overview); 5 μm (detail).
(TIF)

**S4 Fig. CLEM of intracellular STY.** HeLa cells expressing LAMP1-GFP (green) were infected with STY WT expressing mCherry (red). Cells were fixed 7 h p.i. and processed for confocal microscopy (**A**, **C**) and TEM of ultrathin sections (**B**, **D**, **E**, **F**). **A**) An infected cell showing a distinct LAMP1-GFP-positive SIF network was identified, further analyzed, ad is shown in maximum intensity protection (MIP). **B**) TEM overview image of a host cell harboring several salmonellae. **Ci, ii, iii**, **Di, ii**) For correlation, TEM and CLSM images were superimposed. **E**) Detail of individual STY cells in SCV with single membrane (red arrowhead, white arrow). **F**) Detail view of a small section of a double membrane SIF with inner (black arrowhead) and outer (blue arrowhead) membrane. Scale bars: 10 μm (**A**), 7 μm (**B**), 2 μm (**D**), 750 nm (**E**),

200 nm (**F**).
(TIF)

**S5 Fig. CLEM of intracellular SPA.** HeLa cells expressing LAMP1-GFP (green) were infected with SPA WT expressing mCherry (red). Cells were fixed 7 h p.i. and processed for confocal microscopy (**A**, **C**) and TEM of ultrathin sections (**B**, **D**, **E**, **F**, **G**, **H**). **A**) An infected cell showing a distinct LAMP1-GFP-positive SIF network was identified, further analyzed and shown as MIP. **B**) TEM overview image of a host cell harboring several salmonellae (red box). **Ci**, **ii**, **iii**, **Di**, **ii**) For correlation, TEM and CLSM images were superimposed. **H**) Detail of individual SPA cells in SCV with single membrane (red arrowhead). **D**, **F**) Overviews showing SIF distal to the SCV, regions shown in details are indicated by light or dark blue boxes. **E**, **G**) Corresponding detail views of double membrane SIF with inner (black arrowhead) and outer (light blue, dark blue arrowheads) membranes. Scale bars: 10 μm (**A**), 7 μm (**B**), 2 μm (**D**), 250 nm (**E**, **G**), 3 μm (**F**), 500 nm (**H**).
(TIF)

**S6 Fig. CLEM of intracellular STM.** HeLa cells expressing LAMP1-GFP (green) were infected with STM WT expressing mCherry (red). Cells were fixed 7 h p.i. and processed for confocal microscopy (**A**, **C**) and TEM of ultrathin sections (**B**, **D**, **E**, **F**, **G**). **A**) An infected cell showing a distinct LAMP1-GFP-positive SIF network was identified, further analyzed and shown as MIP. **B**) TEM overview image of a host cell harboring several salmonellae. **Ci, ii, iii, Di, ii**) For correlation, TEM and CLSM images were superimposed. **E**) Detail of individual STM cells in SCV with single membrane (red arrowhead, white arrow). **F**, **G**) Detail views of small sections of double membrane SIF with inner (black arrowhead) and outer (blue arrowhead) membrane. Scale bars: 10 μm (**A**), 7 μm (**B**), 2 μm (**D**), 750 nm (**E**), 200 nm (**F**, **G**).
(TIF)

**S7 Fig. Intracellular proliferation of STM and STY determined by flow cytometry.** The single dot plots of the experiment shown in **Fig 8** are displayed. HeLa cells (**A**), RAW264.7 macrophages without IFN-γ-activation (**B**), or U937 macrophages without IFN-γ-activation (**C**) were used as host cells. Gating for infected host cells was made using non-infected cells (grey dots) as control. Overlays of dot plots show WT *Salmonella* at 1 h or 24 h p.i. in red and blue, respectively, and Δ*ssaV* or Δ*ssaK Salmonella* at 1 h or 24 h p.i. in orange and green, respectively. The X-mean values for mCherry fluorescence intensities of infected cells are indicated the gates for infected host cells. Assays were performed in three biological replicates, one representative replicate is shown.
(TIF)

**S8 Fig. Biosynthetic capacity of STM WT and SPI2-T3SS-deficient strains in various host cells.** STM (green) WT (open bars) and SPI2-T3SS-deficient Δ*ssaV* (hatched bars) strains, each harboring plasmid p4928 were used to infect HeLa cells, U937 macrophages without IFN-γ-activation, or RAW264.7 macrophages without IFN-γ-activation as indicated. Infection, induction of expression and analyses by FC were basically performed as described for **Fig 9**. Host cells were lysed at the indicated time points, bacteria-sized particles were gated based on DsRed fluorescence, and the sfGFP-positive population was gated. The means and standard deviations of sfGFP-induced *Salmonella* of three biological replicates are shown. Statistical analyses were performed as described for **Fig 9**.
(TIF)

**S1 Table. Oligonucleotides used in this study.**
(DOCX)

**S2 Table. Antibodies used in this study.**
(DOCX)

## Acknowledgments

We thank Dr. Ohad Gal-Mor for providing STY and SPA strains, and the members of the Sal-HostTrop consortium for fruitful discussions. The generation of reporter plasmids by Ursula Krehe is gratefully acknowledged.

## Author Contributions

**Conceptualization:** Tatjana Reuter, Felix Scharte, Michael Hensel.

**Data curation:** Tatjana Reuter, Felix Scharte, Michael Hensel.

**Formal analysis:** Tatjana Reuter, Felix Scharte, Rico Franzkoch, Viktoria Liss, Michael Hensel.

**Funding acquisition:** Michael Hensel.

**Investigation:** Tatjana Reuter, Felix Scharte, Rico Franzkoch, Viktoria Liss.

**Methodology:** Tatjana Reuter, Felix Scharte, Rico Franzkoch, Viktoria Liss.

**Project administration:** Felix Scharte, Michael Hensel.

**Resources:** Michael Hensel.

**Supervision:** Michael Hensel.

**Validation:** Tatjana Reuter, Felix Scharte, Viktoria Liss.

**Visualization:** Tatjana Reuter, Felix Scharte, Rico Franzkoch, Viktoria Liss.

**Writing – original draft:** Tatjana Reuter, Felix Scharte, Rico Franzkoch, Viktoria Liss, Michael Hensel.

**Writing – review & editing:** Michael Hensel.

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
