## [Decision Letter · Decision Letter 0]

9 Feb 2021

Dear Prof. Dr. Hensel,

Thank you very much for submitting your manuscript "Single cell analyses reveal distinct adaptation of typhoidal and non-typhoidal Salmonella enterica serovars to intracellular lifestyle" for consideration at PLOS Pathogens. As with all papers reviewed by the journal, your manuscript was reviewed by members of the editorial board and by several independent reviewers. In light of the reviews (below this email), we would like to invite the resubmission of a significantly-revised version that takes into account the reviewers' comments.

We cannot make any decision about publication until we have seen the revised manuscript and your response to the reviewers' comments. Your revised manuscript is also likely to be sent to reviewers for further evaluation.

Sincerely,

Andreas J Baumler

Associate Editor

PLOS Pathogens

Brian Coombes

Section Editor

PLOS Pathogens

Kasturi Haldar

Editor-in-Chief

PLOS Pathogens

orcid.org/0000-0001-5065-158X

Michael Malim

Editor-in-Chief

PLOS Pathogens

orcid.org/0000-0002-7699-2064

Reviewer's Responses to Questions

**Part I - Summary**

Reviewer #1: In this work the authors study the intracellular lifestyle of typhoidal Salmonella and compare it with that of nontyphoidal Salmonella Typhimurium, which has served as the model pathogen of foodborne infection. Their findings from the application of diverse microscopy techniques to cell cultures infected with Salmonella Typhi and Paratyphi suggest SPI-2 effector translocation by typhoidal Salmonella upon entry into epithelial cells. Interestingly, fluorescence reporter strains of Salmonella Typhi and Paratyphi reveal these strains to activate SPI-2 virulence genes in only a small subpopulation whereas the remaining intracellular bacteria do not activate this virulence program. This is in apparent contrast to S. Typhimurium for which it is known that a substantially higher fraction of intracellular bacteria activate SPI-2. CFU and flow cytometry assays indicate that S. Typhi deletion mutants devoid of SPI-2 are not compromised in intracellular survival, but show a reduced proliferation compared with wild-type bacteria.

Reviewer #2: Reuter et al describe a study addressing the important question of differences in interaction of host adapted typhoidal Salmonella (TS) serovars with a broad host range non-typhoidal Salmonella (NTS) serovar, Typhimurium. The authors use a series of sophisticated experimental approaches to quantify and compare expression and function of T3SS-2 in the three serovars of Salmonella. They find that although T3SS-2 is active and functional in TSs, the level and distribution of Salmonella in host cells is very much reduced. It is speculated that this is associated with the stealth lifestyle of TS serovars.

Reviewer #3: Typhoidal Salmonella serovars including serovar Typhi (S. Typhi) and Paratyphi A (S. Paratyphi A) causes systemic infection only in human. Mouse infection with serovar Typhimurium (S. Typhimurium) is considered as a good animal model for studying the pathogenesis of Typhoid and Paratyphoid. Previous many reports have demonstrated the crucial contribution of T3SS encoded by SPI-2 (T3SS-2) to dissemination of bacteria into systemic sites. S. Typhi and S. Paratyphi A have similar T3SS-2 gene cluster on SPI-2, however repertoire of T3SS-2 effector is different from that of S. Typhimurium. The roles of T3SS-2 for their pathogenesis remains unclear.

In the manuscript entitled “Single cell analyses reveal distinct adaptation of typhoidal and non-typhoidal Salmonella enterica serovars to intracellular lifestyle” by Reuther T. et al. analyzed expression and functions of T3SS-2 of intracellular typhoidal Salmonella serovars by using the single cell analysis methods adopted from previously established in S. Typhimurium study. In combination of flowcytometric analyses and imaging, the authors indicated only a small part of intracellular typhoidal serovars express T3SS-2 related genes and translocate T3SS-2 effectors. The authors also suggest that low T3SS-2 activation leads less SIF formation and replication in the tissue cultures compared to S. Typhimurium which may play a role in the stealth strategy of the typhoidal Salmonella serovars.

The obtained data is interesting and important to understand the pathogenesis of typhoidal Salmonella serovar, and the techniques used in this study is potentially useful for not only Salmonella but also other intracellular pathogens. However, I would like to point out some concerns and questions as listed below. Addressing the questions and changing manuscript would be better suited for the publication in the journal PLoS Pathogens.

**Part II – Major Issues: Key Experiments Required for Acceptance**

Reviewer #1: The role of SPI-2 for typhoidal Salmonella has indeed been poorly understood. The presented data are sometimes a bit confusing and not easy to follow, even for someone relatively familiar with these infection systems, and data visualization could be improved. What makes it hard to draw general conclusions is that often the serovars behave inconsistently between the different cell models (even between U937 and RAW264.7, which are both macrophage-like cell types). The study reports observations, but falls relatively short on providing any new mechanistic/functional insights. E.g. given their findings that SPI-2 expression is heterogeneous and the proportion of SPI-2+ vs. SPI-2- intracellular bacteria very different for the typhoidal compared with the nontyphoidal serovars, a natural question to ask is what the molecular source of this heterogeneity (or its consequence) could be. The observation that SPI-2-deficient S. Typhi show a reduced intracellular proliferation rate is interesting; yet the relevance for the in vivo setting and for the outcome of disease remain unclear. In my opinion, a mechanistic/functional aspect of this kind would be required to justify publication in this journal.

Reviewer #2: Several of the assays report the distribution of responses of STM, SPA and STY in various cell types in the form of a single representative experiment, of three replicates. Increases or decreases of each Salmonella strain is reported but how representative and the amount of variation between biological replicates is not apparent from the manuscript. Although the conclusions based on the representative data seems very clear and appropriate it is difficult to interpret the reproducibility based on the presentation of a single representative observation of the distribution of a response. This applies to the presentation of data in Figure 1, Figure 6 and Figure 9. This could be addressed by enumerating the response by gating and applying statistical analysis to the biological replicates.

Reviewer #3: 1. The authors tried to quantify the level of T3SS-2 expression (Fig. 1), SCV integrity (Fig. 6), intracellular proliferation (Fig. 8), biosynthetic capability (Fig. 9) of intracellular S. Typhimurium, S. Typhi and S. Paratyphi A by using flowcytometric analyses in order to compare among them. However, the authors don’t repeat any assays. Authors should repeat each assay and make graphs with statistics to strengthen the data.

2. The authors have shown that gentamicin protection assay indicates no contribution of T3SS-2 to replication of S. Typhi in RAW264.7 and U937 cells (Fig. 7). From the graph it’s hard to know either S. Typhi wild-type can’t replicate in the macrophages unlike S. Typhimurium (or rather replicate like S. Typhimurium ssaV) or S. Typhi can replicate intracellularly regardless of the presence of T3SS-2. To clear it, the authors should repeat the experiment Fig 8 and show growth rate (1h/24h) of S. Typhimurium and S. Typhi in each tissue culture beside the flow data plots.

3. The ssaG expression of S. Typhi in RAW264.7 looks better than that of S. Typhimurium (Fig. 1D), but no effect on S. Typhi intracellular replication (Fig. 7). Some explanation should be required for the gap.

4. Translocation of sifA (sifA::M45) by the typhoidal Salmonella serovars is required (Fig. 3) because SIF formation is focused in this study. It remains unclear the reduction of SIF formation observed by infection with S. Typhi and S. Paratyphi A (Fig. 4) is resulted in reduction of sifA expression and translocation.

5. The images in Fig 4A show SIF at 8h post infection, but graphs in Fig. 4B show SIF at 16h and 24h post infection. Authors should the frequency of SIF positive cells in the graph at 8h instead of one of later time points.

6. Some sections don’t have authors’ conclusion from consequence of each result. That may make readers get confused. For instance, in the section “SCV integrity and cytosolic access of intracellular STY and SPA”, author should give any opinion why S. Typhi ∆sifA strain didn’t reduce its SCV integrity unlike S. Typhimurium ∆sifA strain, and how it is relating with other obtained results.

8. Biosynthetic capability of intracellular Salmonella is required for T3SS-2 activity (Fig. 9)? The author concluded the typhoid Salmonella serovars have lower capacity than S. Typhimurium to maintain a biosynthetic active sated over prolonged presence in host cells (lane 327). But it remains unclear how T3SS-2 is involved in the activity. The author should determine if a S. Typhimurium T3SS-2 mutant abolished the activity.

**Part III – Minor Issues: Editorial and Data Presentation Modifications**

Reviewer #1: • Throughout the manuscript, the authors should do a better job explaining the rationale for using specific effector mutants (e.g. delta-ssaV, delta-ssaR, delta-ssaK, etc.).

• Lines 162-163: “The amounts of effector proteins (…) than for STM-infected host cells (data not shown).” Please, do show these data. Similar for the following sentence “Also, the frequency (…) was lower.” This statement seems not to be supported by any of the presented data.

• Lines 168-169: “Translocation of SseJ is possible in STY and SPA, although sseJ is a pseudogene in these serovars (…)” Please elaborate, e.g. which part of the gene is mutated in typhoidal compared with nontyphoidal Salmonella? Would this mutation be expected to prevent transcription/translation or result in the production of a shorter protein? Can the authors provide additional (more convincing) images that this effector is indeed produced and translocated by SPA and STY (there is barely any signal visible in current Fig. 3 for this effector).

• Lines 192-193: “Since imaging (…) not assessed.” Please explain this in more detail as it may not be obvious for all readers why fixation would prevent SIF detection in macrophages.

• Lines 216, 218: The acronyms “CLEM” and “TEM” need to be introduced.

• Line 268: “(…) information and functionally of (…)” Please correct.

• Line 277: “(…) and the deficient in proliferation (…)” Please correct.

• Line 281: The acronym “FP” needs to be explained.

• Line 294-296: “We anticipated that (…)” I am not sure if I understand the authors’ hypothesis. Hasn’t it been shown that in vivo Salmonella persisters retain metabolic activity (see work from Helaine lab)?

• Line 419: “(…) a recently organoid model (…)” Please correct.

• Fig. 1A: The main conclusion the authors draw from these data is that all the tested Salmonella serovars activate SPI-2 in PCN-P (see line 116 in the Results section). What is not mentioned, however, is that the magnitude of SPI-2 induction (intensity of the reporter signal) per bacterial cell is substantially lower for SPA and STY as compared with STM. Thus, it seems that not only the proportion of SPI-2+ bacteria is lower in the typhoidal serovars (panels B-D), but also the expression levels of SPI-2 genes within the SPI-2+ subpopulation. Can the authors please comment?

• Fig. 1D: As judged by eye, the shoulder of the red graph (STY WT) that extends into the “induced” population in the presence of IFNg implies a smaller value than the reported 51.45%. Can the authors comment, please?

• Fig. 3: The effects seem relatively subtle. Can effector translocation please be quantified over multiple cells (e.g. how many of the Salmonella-infected cells were stained positive for the respective effectors?) and these data be plotted here, rather than/in addition to the single (representative) microscopy image? In addition, inclusion of STM in these experiments would be appreciated, as it would allow for direct comparison of the results between all three serovars. (E.g. in regard of my above comment: How would translocation compare between typhoidal (pseudogenized) and nontyphoidal SseJ?)

• Fig. 4A: Please explain in the figure legend what the insets and yellow arrowheads refer to.

• Fig. 4B: The figure legend informs about the statistical analysis (n.s.; *, **). However, not all bars in this panel have any statistical label. Can the authors please add this?

• Fig. 6: As the authors rightfully mention (lines 249-250) STM delta-sifA mutants are deficient in maintaining SCV integrity, resulting in their enhanced cytosolic exposure. As inferred from panel B, it seems as if SifA had the opposite effect in STY inside HeLa cells (where sifA deletion apparently DEcreased cytosolic exposure). Can the authors please comment?

• Fig. 7: Rather than plotting replication relative to the WT, I’d prefer if replication values were plotted relative to the respective 1 h values. This way, differences between the different cell lines and serovars could be deduced.

• Fig. 8A is confusing: It looks as if the delta-ssaV mutant (at 1 h p.i.) had an increased SSC-H signal compared to the WT (i.e., the yellow/orange scatters extend above the red scatters). Additionally, at 24 h the ssaV mutant (green dots) seems to extend to the right (mCherry axis) as compared to the WT (blue cloud) despite the reported numbers suggesting that the mCherry signal of the ssaV mutant is only half the signal of the WT at 24 h. Can the authors comment? I wonder if this representation really is the best way of visualizing the results. Also, Salmonella reporter plasmids have been reported that allow for a more accurate quantification of (intracellular) replication (PMIDs: 20133586; 25126781). The authors may want to switch to one of these systems to measure intracellular proliferation more precisely.

• Fig. 9A: In case of SPA, there seems to be heterogeneity in AHT responsiveness in vitro (which is absent from STM and STY). Is this worth mentioning?

• Fig. 9C, D: For both SPA and STY there seems to be some kind of oscillation in AHT responsiveness. That is, inside both HeLa and U937 cells the induced population was high at 8 h p.i., then dropped at 16 h p.i., and increased again at 24 h p.i. Do the authors have any explanation for this phenomenon?

Reviewer #2: 1. L27. Not all S. Typhimurium are associated with a broad host range and so consider '....many S. enterica serovar Typhimurium (STM)....'.

2. L35. Suggest being precise about which single cell analyses were used in this study, 'such as' suggest these are only examples and makes the reader believe there may be others.

3. L51. Commonly, poor infrastructure for sanitation rather than simply hygiene are implicated in TS infections.

4. L79. Typo 'SP2-T3SS'

5. 82-84. Suggest discussing the humanised mice model described by Karlinsey et al, 2019 here too.

6. L121. Please indicate the type of cells HeLa and U937 are as is done for RAW264.7 cells. Most will know that HeLa are epithelial but should be added anyway. U937 are less familiar.

7. L171. Please specify if these observations are in vivo or in cultures cells and if these were phagocytic or non phagocytic cells , or reported in all?

8. L175. Typo, should be 'completely'?

9. L294. suggest '...capable of protein....’

10. L390. Typo, should be 'subtle'?

11. L531-553. Please describe the statistical analysis of the cytometry data.

Reviewer #3: 1. Author should describe the origin of each tissue culture (lane 122) and mention why those tissue cultures are chosen for the study.

2. Author should explain a reason why only RAW264.7, but not other macrophage cell lines, needed to get primed by IFN-gamma (lane 132). I am also confused whether or not authors used the primed RAW264.7 cells for latter experiments.

3. In human macrophages, all serovars didn’t show expression of ssrB unlike in U937. The authors should give any suggestions for this point why T3SS-2 expression was different in between human macrophage and U937.

4. Lane 192 “Since imaging was…” means SIF is not observed in fixed macrophages infected with Salmonella? Any reference?

5. S. Typhimurium wild type showed reduction of SCV integrity in HeLa but not in U937 over the course of infection (Fig. 6). How about in RAW264.7 cells? I wonder if S. Typhimurium doesn’t reduce the SCV integrity specifically observed in macrophages.

6. Some abbreviations are unknown for (“FC” on lane 259; “FP” on lane 281). Bacterial names in the legends of Figs S4-S6 are not consistent with the bacterial names of images. Lane 711, “in revision”? They are needed to be fixed.

PLOS authors have the option to publish the peer review history of their article (what does this mean?). If published, this will include your full peer review and any attached files.

Reviewer #1: No

Reviewer #2: No

Reviewer #3: No
---

## [Editor Report · Decision Letter 1]

14 May 2021

Dear Prof. Dr. Hensel,

We are pleased to inform you that your manuscript 'Single cell analyses reveal distinct adaptation of typhoidal and non-typhoidal Salmonella enterica serovars to intracellular lifestyle' has been provisionally accepted for publication in PLOS Pathogens.

Best regards,

Andreas J Baumler

Associate Editor

PLOS Pathogens

Brian Coombes

Section Editor

PLOS Pathogens

Kasturi Haldar

Editor-in-Chief

PLOS Pathogens

orcid.org/0000-0001-5065-158X

Michael Malim

Editor-in-Chief

PLOS Pathogens

orcid.org/0000-0002-7699-2064
---

## [Editor Report · Acceptance letter]

14 Jun 2021

Dear Prof. Dr. Hensel,

We are delighted to inform you that your manuscript, "Single cell analyses reveal distinct adaptation of typhoidal and non-typhoidal Salmonella enterica serovars to intracellular lifestyle," has been formally accepted for publication in PLOS Pathogens.

Best regards,

Kasturi Haldar

Editor-in-Chief

PLOS Pathogens

orcid.org/0000-0001-5065-158X

Michael Malim

Editor-in-Chief

PLOS Pathogens

orcid.org/0000-0002-7699-2064